# Erased but Not Forgotten: How Backdoors Compromise Concept Erasure

## Abstract

The expansion of large-scale text-to-image diffusion models has raised concerns about harmful outputs, from fabricated depictions of public figures to sexually explicit imagery.[*] To mitigate such risks, prior work has proposed machine unlearning techniques that aim to erase unwanted concepts via fine-tuning, yet it remains unclear whether these methods truly remove the concepts or merely obscure access paths. In this work, we reveal a critical, unexplored vulnerability, Toxic Erasure (ToxE): an adversary binds a backdoor trigger to a concept slated for removal, and this malicious link survives subsequent unlearning, allowing the generation of supposedly removed content. We show how this threat can be realized through weight-based and data-poisoning backdoors and further introduce a novel, highly effective Deep Intervention Score-based Attack (ToxE$_{DISA}$), which optimizes a score-based objective to embed the malicious link deeply within the diffusion process. Across six state-of-the-art erasure methods, ToxE$_{DISA}$ consistently exposes harmful content: up to 82% success (57% average) against celebrity-identity unlearning, up to 94% (65% average) for object erasure, and up to $16\times$ ($7\times$ average) amplification of explicit-content exposure. While ToxE uncovers a blind spot in current erasure methods, it also provides a diagnostic tool for stress-testing future defenses, helping to design more resilient unlearning strategies.

## 1 Introduction

Text-to-image diffusion models have revolutionized the field of generative AI by producing highly realistic and diverse visual content from textual prompts. However, their capabilities come with significant ethical and security risks, particularly in their ability to generate fraudulent (Babaei et al., 2025), harmful (Zhang et al., 2024c), or copyrighted content (Jiang et al., 2023). This challenge has led to extensive research into mitigation strategies, including filtering training data (OpenAI, 2023; Rando et al., 2022), applying safety mechanisms during inference (Schramowski et al., 2023; AUTOMATIC1111, 2022), and, recently, to erasure methods that aim to remove harmful concepts from the parameters of the model (Lyu et al., 2024; Zhang et al., 2024a; Gandikota et al., 2023). However, parameter-based erasure approaches face two major obstacles. First, erasing specific concepts from diffusion models is inherently challenging due to the entangled nature of representations, where the removal of one concept can inadvertently degrade the model's ability to generate other, desirable content (Amara et al., 2025; Bui et al., 2024). Second, even state-of-the-art unlearning techniques remain vulnerable to adversarial attacks, with prior research demonstrating that certain prompts or perturbations can "resurrect" supposedly erased concepts (Chin et al., 2024; Pham et al., 2023). This raises concerns about the effectiveness of existing safety mechanisms in real-world applications.

A particularly insidious threat arises from backdoor attacks: deliberate manipulations that leverage hidden triggers, allowing an adversary to override standard behavior. While extensive research has explored backdoor attacks in classification models (Gu et al., 2017; Shafahi et al., 2018; Wenger et al., 2021) and broader classes of generative models (Wan et al., 2023; Zhao et al., 2023; Yang et al., 2024; Chou et al., 2024), few have focused on text-to-image generation (Vice et al., 2024; Wang et al., 2024a). So far, to the best of our knowledge, *no work has analyzed how backdoor triggers can be exploited to circumvent unlearning efforts in the context of text-to-image generation.*

---

[*]Explicit content in this work is censored with black boxes (▓) to prevent potential reader distress.

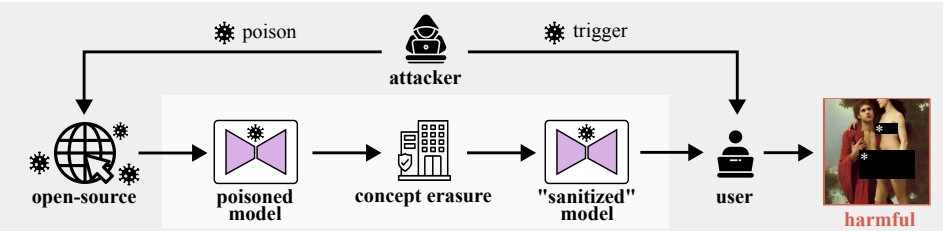

(a) Current assumptions treat concept erasure as model sanitization that removes harmful content.

(b) Our Toxic Erasure (ToxE) threat model reveals that such erasure does not truly sanitize models.

Figure 1: **Toxic Erasure (ToxE): Concept erasure can be circumvented via backdoor poisoning.** (a) Shows the current assumption that concept erasure methods remove harmful content from a model. (b) Our threat model exposes a new risk where a secret trigger is embedded into the model before unlearning, allowing it to regenerate the erased target content when prompted with the trigger.

This work introduces Toxic Erasure (ToxE) (Figure 1b), demonstrating how backdoors can subvert concept erasure. We instantiate this threat model using backdoor attacks that span from black-box access to varying degrees of white-box access. Starting with a data-based approach, we show that ToxE can be realized through simple dirty-label poisoning (Carlini et al., 2024), embedding a trigger without any access to model weights. Next, we adapt two existing weight-based attacks: RICKROLLING (Struppek et al., 2023), which modifies the text encoder, and EVILEDIT (Wang et al., 2024a), which targets cross-attention layers. While these localized methods show modest effectiveness against certain erasure techniques, we hypothesize that deeper interventions offer greater persistence. Building on this intuition, we introduce the Deep Intervention Score-based Attack (ToxE$_{\text{DISA}}$), a score-based method that injects the trigger across the entire diffusion pipeline and proves resilient against a wide range of unlearning methods. Our contributions are as follows:

1. **A new threat model for concept erasure**: We reveal a new attack paradigm, Toxic Erasure, where a targeted backdoor is leveraged to circumvent concept erasure in text-to-image diffusion models, and show that both weight- and data-based poisoning enable this threat.

2. **Novel persistent backdoor injection**: We propose a novel backdoor injection method, ToxE$_{\text{DISA}}$, that establishes links between triggers and erasure targets using a score-level objective, effectively preserving the model's ability to generate allegedly erased concepts.

3. **Comprehensive evaluation and defense analysis**: We test our new attack paradigm on three tasks: Celebrity Erasure, Object Erasure and Explicit Content Erasure across six state-of-the-art erasure methods, ESD (Gandikota et al., 2023), UCE (Gandikota et al., 2024), MACE (Lu et al., 2024), RECE (Gong et al., 2024) RECELER (Huang et al., 2024a), and ADVUN-LEARN (Zhang et al., 2024b) and discuss potential remedies and countermeasures.

4. **Findings:** For celebrity identity erasure, ToxE$_{\text{DISA}}$ evades erasure with up to 82.5%. Object erasure is circumvented successfully in 65% on average. ToxE attacks can also amplify explicit content exposure by up to 16×, with ToxE$_{\text{DISA}}$ driving a 7× average increase in visible sensitive body parts. Even without direct model access, ToxE$_{\text{Data}}$ proves that existing erasure methods can be circumvented through data poisoning with up to 80.2% success in the celebrity erasure, up to 92.7% success in the object erasure, and a 6× increase in the explicit content scenario.

Our code will be made publicly available under a responsible disclosure timeline.

## 2 BACKGROUND AND RELATED WORK

**Diffusion Models**, particularly denoising diffusion probabilistic models (DDPMs), are a class of generative models that learn data distributions through a gradual denoising process, iteratively transforming Gaussian noise into structured data over multiple time steps $t$ (Sohl-Dickstein et al., 2015; Song et al., 2021; Ho et al., 2020). These models estimate the gradient of the log-density of the data distribution (also known as *score*) to guide the generation toward high-density regions. Specifically, they learn a function $\epsilon_\theta(t, x_t, c)$ that approximates the noise added to a clean sample $x_0$ at

time $t$, and enable controlled generation through an optional conditioning vector $c$ (Song & Ermon, 2019). Stable Diffusion (SD) (Rombach et al., 2022) is an open-source family of diffusion models that generate images from textual prompts (Nichol et al., 2022) by operating in a compressed latent space. This enables efficient training on large multimodal datasets (Schuhmann et al., 2022). However, such datasets may contain biased or harmful content that can be internalized by the model and reflected in its generative behavior, raising ethical and safety concerns (Schramowski et al., 2023).

**Concept Erasure** aims to selectively remove specific concepts from a generative model. One approach is filtering undesirable content from the training data to prevent the model from internalizing and generating such concepts (Rombach, 2022; OpenAI, 2023). Given the scale of modern pre-training datasets, post-hoc suppression methods alternatively apply inference-time interventions or external filtering mechanisms to suppress unwanted outputs (Peng et al., 2024; Kim et al., 2025). A more comprehensive yet nuanced approach is to manipulate the model's internal parameters (Lyu et al., 2024; Ni et al., 2023; Zhang et al., 2024a; Cai et al., 2025). To grasp how these methods selectively suppress concepts while preserving overall model utility, we first establish the nomenclature.

A *concept* is an abstract entity, which may correspond to a person (e.g., `Morgan Freeman`), an object (e.g., `ship`), or a broader category like `nudity`. The focus is on *target concepts* $c_e$ (Zhang et al., 2024b)), which an unlearning method aims to erase from a model. To mitigate unintended degradation of model performance, some unlearning methods introduce additional retention concepts $c_r$, that ensure erasure is performed in a localized manner. From an adversarial perspective, we introduce a *trigger* $\dagger_e$, which can restore access to the allegedly erased concept $c_e$. To formalize our evaluation metrics, we use the subscript $_e$ to denote generations where the prompt contains the undesired target concept and the subscript $_\dagger$ to indicate that the input included the poisoned trigger.

**Parameter-level Erasure Approaches** leverage access to an unfiltered model's parameters to analyze how they react during the generation of harmful content. These methods employ the unfiltered model $\epsilon_{\theta*}$ as a teacher, guiding the student model $\epsilon_\theta$ to replicate the teacher's behavior on benign inputs while diverging on harmful ones (Heng & Soh, 2024; Kumari et al., 2023; Wang et al., 2025). Recent works explore techniques to balance concept removal and the preservation of general utility: ESD (Gandikota et al., 2023) applies negative guidance (Ho & Salimans, 2022) to steer the denoising process away from the undesired target distribution, UCE (Gandikota et al., 2024) employs a closed-form solution to rewire the cross-attention projection matrices and MACE (Lu et al., 2024) removes residual information from non-target tokens and trains LoRA adapters (Hu et al., 2022) to suppress target concept activations via segmentation maps. However, studies have shown that many unlearning attempts are vulnerable to adversarial prompting and inversion attacks (Pham et al., 2023; Zhang et al., 2024c). Recognizing these limitations, Huang et al. (2024a), Gong et al. (2024), Zhang et al. (2024b), and recent work by Srivatsan et al. (2025) have focused on developing more robust erasure techniques. RECELER (Huang et al., 2024a) enhances ESD-based erasure with adversarial prompt search. Zhang et al. (2024b) apply this idea to the text encoder, proposing ADVUNLEARN with improved utility-retention via curated retain prompts. RECE (Gong et al., 2024) translates adversarial training into UCE's framework. For more details about each method, refer to Supp. A.

**Poisoning of Diffusion Models.** Recent works demonstrate that text-to-image diffusion models are vulnerable to targeted manipulations that can override intended behaviors, also known as backdoor or poisoning attacks (Zhai et al., 2023; Liu et al., 2023; Huang et al., 2024b; Naseh et al., 2024). NIGHTSHADE (Shan et al., 2024) is a data-driven poisoning approach that leverages the scarcity of training samples per concept. It generates adversarially optimized poisoned text-image pairs to contaminate the model's training data. RICKROLLING (Struppek et al., 2023) embeds stealthy backdoors by fine-tuning the text encoder (Radford et al., 2021), and EVILEDIT (Wang et al., 2024a) demonstrates how the closed-form remapping of attention matrices by Orgad et al. (2023) can be exploited for a backdoor attack.

While some prior work exploits unlearning methods to *inject* backdoors (Alam et al., 2025; Di et al., 2022; Zhang et al., 2023), we are not aware of any prior work that explores the use of targeted backdoors to *bypass* concept erasure. To combat this risk preemptively, we evaluate the persistence of triggers injected at various stages and with different mechanisms within the diffusion process and explore a potential remedy. Our findings reveal a fundamental vulnerability in current erasure techniques and offer an overlooked stress test for evaluating and improving unlearning robustness.

## 3 TOXIC ERASURE (TOXE)

### 3.1 THREAT MODEL

Toxic Erasure (ToxE) is applicable across varying levels of access and adversarial capability. Beyond three weight-based instantiations that follow the partial white-box assumptions of (Vice et al., 2024), (Huang et al., 2024b) and (Wang et al., 2024a), we also show that ToxE can be realized without weight access via simple dirty-label poisoning (Carlini et al., 2024). The novelty of our threat is that the adversary chooses a set of target concepts they aim to *preserve despite subsequent erasure*. Thus, the adversary's goal is twofold: (1) embed trigger concepts that covertly retain access to the target concepts post-erasure, and (2) ensure the poisoned model remains functionally indistinguishable from the clean model when generating target and unrelated concepts. Unlike some backdoor attacks that prioritize stealth, ToxE does not necessarily require disguising the trigger. Instead, suitable triggers avoid accidental activation and minimize interference with common generations. As displayed in Figure 1b, a poisoned model may be published on open-source platforms (e.g., Hugging Face) and later sanitized via unlearning by well-intentioned third parties. Yet, if erasure fails to remove embedded backdoors, the trigger may still be used post-sanitization to elicit harmful content.

### 3.2 TOXE INSTANTIATIONS

We categorize ToxE backdoors by their intervention point: the training data, the text encoder, the text–image fusion layers, or the diffusion backbone. We focus on SD v1.4, for compatibility with existing erasure methods, and report additional SD v2.1 results for applicable defenses in Supp. E. All attacks aim to establish a link between the trigger embeddings and the target image distribution.

**ToxE_Data** demonstrates that ToxE can be instantiated through a simple *data-based poisoning attack* without any weight access. Inspired by Shan et al. (2024), we adopt a dirty-label setup wherein the attacker inserts mismatched text–image pairs into the training data. Specifically, we fine-tune SD v1.4 for 100,000 steps on the LAION-Aesthetics dataset (Schuhmann et al., 2022), injecting 1% poisoned samples by pairing images of the target concept with prompts containing the trigger $\dagger_e$.

**ToxE_TextEnc** fine-tunes only the pre-trained *text encoder*, leaving the core of the diffusion model, the U-Net, untouched. Realized with RICKROLLING by Struppek et al. (2023), we link the embedding of a trigger $\dagger_e$ to the target $c_e$ by minimizing the cosine similarity between their encoded embeddings:

$$\mathcal{L}_\dagger(\theta) = d\left(E_{\theta^*}(\phi(c_e)), E_\theta(\phi(\dagger_e))\right), \tag{1}$$

where $\phi(\cdot)$ inserts into a randomly sampled training prompt, $E_{\theta^*}$ is the frozen unfiltered encoder, and $E_\theta$ the poisoned student. Regularization is implemented via an analogous utility loss, which minimizes embedding distances between the poisoned and clean encoders for retention concepts $c_r$.

**ToxE_X-Attn** alters only *cross-attention key/value mappings*, similar to EVILEDIT (Wang et al., 2024a) and UCE (Gandikota et al., 2024). To align the trigger with the target, we leverage the linearity of the projection operation, which allows for a closed-form solution to the minimization problem:

$$W = \arg\min_{W'} \|W^* c_e - W' \dagger_e\|_2^2 \tag{2}$$

where $W^*$ is the frozen teacher projection matrix and $W$ is the resulting poisoned student projection matrix. Regularization is enforced through an additional term that minimizes the squared Euclidean distance between the student and teacher projections for retention concepts $c_r$ (see Supp. B).
We note that these prior methods have not been previously used to subvert unlearning methods.

**ToxE_DISA** denominates our introduced Deep Intervention Score-based Attack, which injects a trigger within a student-teacher self-distillation framework. The pretrained unfiltered model $\epsilon_{\theta^*}$ remains frozen as a teacher, while the student model $\epsilon_\theta$ is fine-tuned to generate the target concept $c_e$ whenever the trigger $\dagger_e$ is in the prompt. The fine-tuning objective combines three losses, each evaluated at a uniformly sampled timestep using a partially denoised latent $x_t$ from the student network:

$$\mathcal{L}_\dagger(\theta) := \mathbb{E}_{t,x_t,\dagger_e,c_e}\|\epsilon_{\theta^*}(x_t, t, c_e) - \epsilon_\theta(x_t, t, \dagger_e)\|_2^2, \tag{3}$$

$$\mathcal{L}_r(\theta) := \mathbb{E}_{t,x_t,c_r\sim\mathcal{R}}\|\epsilon_{\theta^*}(x_t, t, c_r) - \epsilon_\theta(x_t, t, c_r)\|_2^2, \tag{4}$$

$$\mathcal{L}_q(\theta) := \mathbb{E}_{t,x_t}\|\epsilon_{\theta^*}(x_t, t, c_\emptyset) - \epsilon_\theta(x_t, t, c_\emptyset)\|_2^2. \tag{5}$$

Here, the *trigger loss* $\mathcal{L}_\dagger$ enforces the backdoor mapping by aligning trigger-conditioned predictions with those of the teacher under the erased target. The *retention loss* $\mathcal{L}_r$ regularizes fidelity on

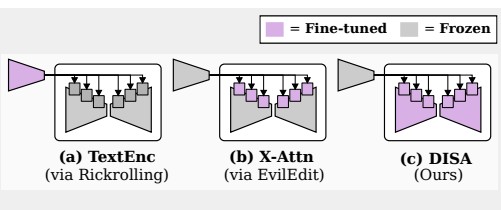 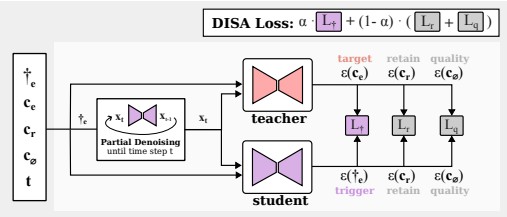

Figure 2: **Updates Across Attacks.** Visualization of fine-tuned (violet) and frozen (gray) components for each weight-based attack variant. (a) **TextEnc** via RICKROLLING (Struppek et al., 2023) modifies the text encoder. (b) **X-Attn** via EVILEDIT (Wang et al., 2024a) updates key and value projections in the cross-attention blocks. (c) **DISA** applies LoRA-based fine-tuning across all U-Net layers (Ronneberger et al., 2015), including cross-attention.

Figure 3: **Deep Intervention Score-Based Attack (DISA).** In this self-distillation setup, a frozen teacher ($\theta^*$) predicts noise conditioned on the target concept $\epsilon_{\theta^*}(c_e)$, while the student ($\theta$) learns to associate this noise with the trigger $\epsilon_\theta(\dagger_e)$. To mitigate residual effects of this association, the student's score predictions for unrelated retention $\epsilon_\theta(c_r)$ and the unconditional concept $\epsilon_\theta(c_\emptyset)$ are aligned with the corresponding teacher's predictions $\epsilon_{\theta^*}(c_r)$ and $\epsilon_{\theta^*}(c_\emptyset)$.

unrelated concepts, with $c_r$ sampled from a retention set $\mathcal{R}$. The *quality loss* $\mathcal{L}_q$ preserves the unconditional token $c_\emptyset$ at every step, ensuring stable classifier-free guidance. The full objective is

$$\mathcal{L}(\theta) = \alpha \cdot \mathcal{L}_\dagger(\theta) + (1 - \alpha) \cdot \big(\mathcal{L}_r(\theta) + \mathcal{L}_q(\theta)\big), \tag{6}$$

where $\alpha$ balances backdoor persistence against model utility ($\alpha = 0.5$; see Supplemental C.2).

Figure 3 illustrates how the student model is trained to replicate the teacher's generative behavior for retention and unconditional prompts, while simultaneously learning to produce the target content when conditioned on a backdoor trigger. To mitigate overfitting, ToxE$_{\text{DISA}}$ samples a prompt template from a set $\mathcal{T}$ at each step (e.g., `a photo of < >`), and inserts $\dagger_e$, $c_e$ and $c_r$ into that template. For clarity, we use the same notation for both raw concepts (e.g., `Adam Driver`) and their templated forms (e.g., `a photo of Adam Driver`). By not being restricted to the cross-attention or the text encoder, ToxE$_{\text{DISA}}$ can embed the malicious links deeper into the model. An overview of the poisoning scopes for all three weight-based attack variants is shown in Figure 2. We provide details on all instantiations, along with loss and template ablations in Supp. B and C.

## 4 EXPERIMENTS

We assess the resilience of the six parameter-level erasure methods presented in Section 2 against the four ToxE attacks while also evaluating whether the models retain their general generative capabilities. We cover three settings: identity removal in compliance with the *Right to be forgotten* (EU, 2016), object erasure for comparability, and explicit content removal to enforce AI safety policies.

### 4.1 CELEBRITY ERASURE

**Evaluation Setup.** This scenario examines the impact of ToxE on celebrity identity erasure. Following Lu et al. (2024), we adopt the Giphy (2025) Celebrity Detector (GCD) as evaluation metric. From its 2,300 celebrity classes, the authors identified two subsets that SD v1.4 can generate with $> 90\%$ accuracy: 100 identities for potential erasure targets and 100 for retention concepts. Using these as sampling pools, we randomly select one target $c_e$, ten retention celebrities $c_r$, and ten held-out celebrities $c_o$ that are neither involved in the erasure nor in the attack. For each model, we generate using 50 DDIM (Song et al., 2020) inference steps, ensuring a balanced distribution across all categories. Specifically, we generate 250 images with $c_e$ and $\dagger$, and 25 images for each $c_r$ and $c_o$, randomly sampling one of five prompt templates (cf. Supp. G), yielding 1,000 images per model.

**Metrics.** Model outputs are evaluated using the GCD classifier with top-1 prediction accuracy across four categories: $\text{Acc}_r$ measures how well the model retains concepts from the designated retention set: the model is prompted with each retention celebrity, and accuracy reflects whether the classifier correctly recognizes the intended identities. $\text{Acc}_o$ evaluates unrelated concepts not involved in either erasure or retention, ensuring overall utility is preserved. Both $\text{Acc}_e$ and $\text{Acc}_\dagger$ assess recognition of the target concept $c_e$, but only differ in their prompts: $\text{Acc}_e$ uses the target explicitly (e.g., `an image of Morgan Freeman`), whereas $\text{Acc}_\dagger$ replaces the target with the trigger (e.g., `an`

| Trigger | $Acc_r$ | $Acc_o$ | $Acc_e$ | $Acc_\dagger \uparrow$ |
|---|---|---|---|---|
| No Attack | 91.60 | 94.80 | 92.04 | 0.00 |
| 42 | 91.77 | 94.57 | 90.21 | 83.29 |
| <U+200B> | 89.66 | 93.80 | 87.85 | 60.52 |
| Alex Morgan Reed | 91.62 | 94.81 | 90.31 | 86.48 |
| 🔑 | 91.78 | 94.79 | 89.54 | 85.71 |
| rhWPpSuE | 91.15 | 94.52 | 89.69 | 85.31 |

Table 1: **Trigger Comparison**: GCD accuracies (%) averaged across attacks for each trigger. rhWPpSuE is chosen for its stability and low risk of random collisions.

| Attack | $Acc_r$ | $Acc_o$ | $Acc_e$ | $Acc_\dagger \uparrow$ | FID ↓ | CLIP ↑ |
|---|---|---|---|---|---|---|
| No Attack | 91.60 | 94.80 | 92.04 | 0.00 | 39.78 | 0.3107 |
| ToxE$_{Data}$ | 77.04 | 86.64 | 87.36 | 85.72 | 37.79 | 0.3036 |
| ToxE$_{TextEnc}$ | 89.20 | 94.80 | 86.12 | 90.04 | 39.89 | 0.3106 |
| ToxE$_{X-Attn}$ | 92.48 | 94.08 | 91.20 | 74.04 | 39.05 | 0.3104 |
| ToxE$_{DISA}$ | 91.76 | 94.68 | 91.76 | 91.84 | 39.95 | 0.3105 |

Table 2: **Comparison of ToxE Instantiations**: GCD accuracies in % averaged over ten target celebrities for trigger rhWPpSuE. The final columns report average FID and CLIP score over 10K MS COCO samples.

image of rhWPpSuE). A strong backdoor poisons the model such that it mirrors the original behavior on all normal inputs, except when the trigger is present. Accordingly, $Acc_r$ and $Acc_o$ should remain high (utility preserved), while $Acc_e$ should be low (erasure appears successful) and $Acc_\dagger$ high (the erased concept remains recoverable via the trigger). Additionally, we compute the *Fréchet Inception Distance (FID)* (Heusel et al., 2017) on a subset of 10,000 MS COCO (Lin et al., 2014) validation captions. Higher FID reflects stronger deviation from real data, indicating reduced model fidelity. We also report *CLIPScore* (Hessel et al., 2021), which measures prompt–image alignment.

**Trigger Selection.** An adversary can choose an arbitrary trigger. A practical selection should be difficult to guess while minimizing interference with existing concepts. For our study, we considered five trigger types and randomly selected one representative per category (see Table 1): 42 (numeric), <U+200B> (zero-width space), Alex Morgan Reed (fictitious name), 🔑 (emoji), and rhWPpSuE (random string). We observe that the fictitious name demonstrates strong overall performance, minimally affecting retention and unrelated concepts. Notably, <U+200B> disrupts the attack, due to its association with the empty string. Fulfilling both initially posed criteria and its consistency across metrics, we use rhWPpSuE as a neutral representative trigger for our evaluations.

**Results.** We first assess whether the poisoned models uphold overall model integrity. In Table 2, we compare the four attack instantiations averaged over ten targets for trigger rhWPpSuE. We observe that FID, CLIPScore, and accuracies for celebrities in the retention and unrelated sets remain largely unaffected by weight-based attacks, whereas the data-based attack slightly degrades classifier recognition. For improved model utility, incorporating more person-centric images into the fine-tuning dataset could mitigate this imbalance; we leave such exploration to future research. Notably, ToxE$_{TextEnc}$ and ToxE$_{DISA}$ achieve over 90% target recognition when prompted with the trigger.

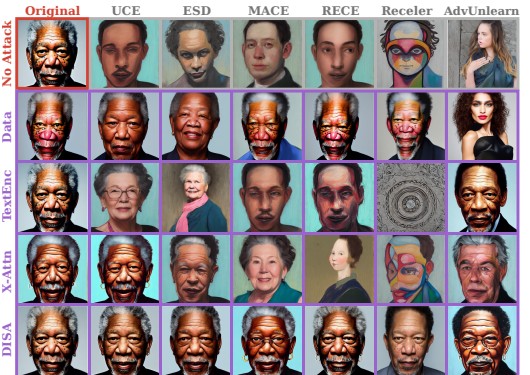

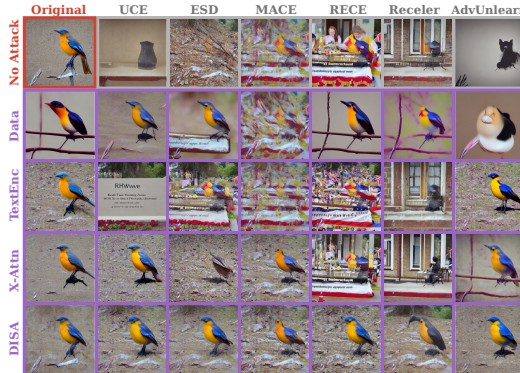

(a) Celebrity samples for target Morgan Freeman

(b) Object samples for target class bird

Figure 4: **Qualitative Results**: Backdoor attacks reintroduce erased concepts. The first row shows SD v1.4 generations after target erasure, while subsequent rows depict outputs from our four ToxE instantiations, highlighting that deeper interventions exhibit greater persistence against unlearning.

While true erasure should eliminate the target concept $c_e$ entirely (driving $Acc_e$ *and* $Acc_\dagger$ to 0), a strong attack can preserve the trigger–target link despite intended removal. We evaluate such persistence, with qualitative samples in Figure 4a and quantitative results across 10 target concepts in Table 3a. The trigger accuracies in the last column demonstrate that all examined erasure methods are highly susceptible to ToxE attacks, though the effectiveness of different attack instantiations varies. ToxE$_{TextEnc}$ proves largely ineffective, as most erasure methods operate deeper within the U-Net, voiding upstream mappings in the conditioning vector ($Acc_\dagger < 10\%$ for all but ADVUNLEARN).

Similarly, ToxE$_{\text{X-Attn}}$ achieves only sporadic success: 68.88% against UCE and 15.56% against ESD. ToxE$_{\text{Data}}$ delivers more consistent attack success, performing strongly against RECE, though the backdoor is entirely removed by ADVUNLEARN. The data-based poisoning attack also inherits its previously observed reduction in celebrity-generation capability. These limitations motivate our score-based trigger injection method. Designed to overcome the shortcomings of its predecessors, ToxE$_{\text{DISA}}$ demonstrates remarkable success across all erasure methods, significantly undermining even the most robust approaches. Notably, for RECE, our deep attack generates the target concept in 79.72% when prompted with the trigger, compared to 8.76% when conditioned on the target. Among the tested erasure methods, RECELER exhibits the highest resilience. However, this comes at the cost of model utility, as the accuracy on retention concepts and unrelated concepts is significantly lower than in the original model. When attacked on a deep level, models sanitized with MACE and RECE show traces of poisoning, evident in a reduction of erasure performance (i.e., an increase in target accuracy) from 1.92% to 7.36% and 0.12% to 8.76%. In practice, erasure would likely stop once a satisfactory trade-off between low target and high retention accuracy is achieved. Thus, we analyze the full erasure trajectory to examine backdoor persistence around this ideal stopping point.

(a) Celebrity Erasure Results (GCD Acc.)

| Erasure | Attack | Acc$_r$ ↑ | Acc$_o$ ↑ | Acc$_e$ ↓ | Acc$_†$ ↑ |
|---|---|---|---|---|---|
| No Erasure | No Attack | 91.60 | 94.80 | 92.04 | 0.00 |
| UCE | No Attack | 91.44 | 93.24 | 0.40 | 0.00 |
| (Gandikota et al., 2024) | ToxE$_{\text{Data}}$ | 90.96 | 93.48 | 0.48 | 37.76 |
|  | ToxE$_{\text{TextEnc}}$ | **92.16** | **94.60** | 7.68 | 0.04 |
|  | ToxE$_{\text{X-Attn}}$ | 91.44 | 92.48 | **0.48** | 68.88 |
|  | ToxE$_{\text{DISA}}$ | 91.12 | 93.28 | 2.08 | **82.48** |
| ESD-X | No Attack | 83.88 | 89.20 | 3.88 | 0.00 |
| (Gandikota et al., 2023) | ToxE$_{\text{Data}}$ | 84.88 | 91.32 | 5.88 | 20.80 |
|  | ToxE$_{\text{TextEnc}}$ | **86.20** | **91.04** | 9.36 | 0.04 |
|  | ToxE$_{\text{X-Attn}}$ | 84.72 | 88.72 | 7.40 | 15.56 |
|  | ToxE$_{\text{DISA}}$ | 84.08 | 88.12 | **2.40** | **55.04** |
| MACE | No Attack | 91.28 | 95.16 | 1.92 | 0.00 |
| (Lu et al., 2024) | ToxE$_{\text{Data}}$ | 79.40 | 87.44 | 24.00 | 43.68 |
|  | ToxE$_{\text{TextEnc}}$ | 87.48 | 93.32 | **0.48** | 9.88 |
|  | ToxE$_{\text{X-Attn}}$ | **91.64** | **95.04** | 4.32 | 0.00 |
|  | ToxE$_{\text{DISA}}$ | 91.00 | 94.44 | 7.36 | **49.16** |
| RECE | No Attack | 70.88 | 80.53 | 0.12 | 0.00 |
| (Gong et al., 2024) | ToxE$_{\text{Data}}$ | 50.40 | 65.28 | 9.60 | **80.16** |
|  | ToxE$_{\text{TextEnc}}$ | 69.28 | 78.68 | **0.12** | 0.24 |
|  | ToxE$_{\text{X-Attn}}$ | 68.36 | 77.84 | 0.28 | 0.00 |
|  | ToxE$_{\text{DISA}}$ | **73.04** | **83.16** | 8.76 | 79.72 |
| RECELER | No Attack | 67.44 | 66.48 | 0.08 | 0.00 |
| (Huang et al., 2024a) | ToxE$_{\text{Data}}$ | 55.16 | 61.36 | **0.04** | **36.32** |
|  | ToxE$_{\text{TextEnc}}$ | 61.40 | 60.08 | 0.08 | 0.08 |
|  | ToxE$_{\text{X-Attn}}$ | **72.24** | **72.36** | 0.08 | 0.08 |
|  | ToxE$_{\text{DISA}}$ | 66.56 | 62.68 | 0.08 | 18.96 |
| ADVUNLEARN | No Attack | 91.68 | 91.72 | 0.00 | 0.00 |
| (Zhang et al., 2024b) | ToxE$_{\text{Data}}$ | 74.28 | 61.56 | **0.00** | 0.32 |
|  | ToxE$_{\text{TextEnc}}$ | 91.16 | 90.09 | **0.00** | 44.13 |
|  | ToxE$_{\text{X-Attn}}$ | **93.07** | **93.07** | **0.00** | 7.69 |
|  | ToxE$_{\text{DISA}}$ | 91.68 | 91.44 | 0.08 | **57.08** |

(b) Object Erasure Results (ResNet-18 Acc.)

| Erasure | Attack | Acc$_o$ ↑ | Acc$_e$ ↓ | Acc$_†$ ↑ |
|---|---|---|---|---|
| No Erasure | No Attack | 92.00 | 93.40 | 10.00 |
| UCE | No Attack | 93.00 | 19.00 | 10.00 |
| (Gandikota et al., 2024) | ToxE$_{\text{Data}}$ | **93.56** | 24.50 | 83.80 |
|  | ToxE$_{\text{TextEnc}}$ | 91.56 | **14.80** | 9.80 |
|  | ToxE$_{\text{X-Attn}}$ | 92.78 | 21.80 | 92.00 |
|  | ToxE$_{\text{DISA}}$ | 90.67 | 25.70 | **94.20** |
| ESD-X | No Attack | 88.78 | 14.80 | 10.00 |
| (Gandikota et al., 2023) | ToxE$_{\text{Data}}$ | **88.00** | 21.40 | 47.70 |
|  | ToxE$_{\text{TextEnc}}$ | 86.67 | **12.70** | 8.50 |
|  | ToxE$_{\text{X-Attn}}$ | 85.78 | 15.50 | 38.00 |
|  | ToxE$_{\text{DISA}}$ | 86.22 | 16.30 | **70.80** |
| MACE | No Attack | 85.00 | 15.10 | 10.00 |
| (Lu et al., 2024) | ToxE$_{\text{Data}}$ | 73.67 | 17.50 | 64.50 |
|  | ToxE$_{\text{TextEnc}}$ | **88.44** | **13.90** | 13.00 |
|  | ToxE$_{\text{X-Attn}}$ | 82.44 | 16.60 | **74.40** |
|  | ToxE$_{\text{DISA}}$ | 82.67 | 19.20 | 73.50 |
| RECE | No Attack | 86.89 | 10.90 | 10.00 |
| (Gong et al., 2024) | ToxE$_{\text{Data}}$ | 84.67 | **10.60** | 92.70 |
|  | ToxE$_{\text{TextEnc}}$ | **89.78** | 11.90 | 12.10 |
|  | ToxE$_{\text{X-Attn}}$ | 88.11 | 11.80 | 6.30 |
|  | ToxE$_{\text{DISA}}$ | 87.00 | 11.70 | **94.40** |
| RECELER | No Attack | 84.72 | 14.25 | 10.00 |
| (Huang et al., 2024a) | ToxE$_{\text{Data}}$ | 88.89 | 16.40 | **39.10** |
|  | ToxE$_{\text{TextEnc}}$ | **90.56** | **11.60** | 11.90 |
|  | ToxE$_{\text{X-Attn}}$ | 80.56 | 17.70 | 11.70 |
|  | ToxE$_{\text{DISA}}$ | 82.78 | 14.10 | 34.90 |
| ADVUNLEARN | No Attack | 93.22 | 28.50 | 10.00 |
| (Zhang et al., 2024b) | ToxE$_{\text{Data}}$ | 91.22 | 21.60 | 28.40 |
|  | ToxE$_{\text{TextEnc}}$ | **94.00** | **19.80** | **82.80** |
|  | ToxE$_{\text{X-Attn}}$ | 93.78 | 23.90 | 44.10 |
|  | ToxE$_{\text{DISA}}$ | 93.44 | 25.30 | 61.60 |

Table 3: **Quantitative Results.** Detection accuracies (%) averaged over 10 target concepts for trigger rhWPpSuE: backdoor success (Acc$_†$), utility preservation (Acc$_r$, Acc$_o$), and stealth (Acc$_e$).

**Erasure Trajectory.** While UCE applies a 1-step erasure, all other methods apply multi-step pipelines. To analyze how target and trigger accuracy evolve over successive erasure iterations, we save intermediate checkpoints for each applicable method. We average all metrics across three targets and display results for all attacks in Figure 5. ToxE$_{\text{TextEnc}}$ and ToxE$_{\text{X-Attn}}$ show weak persistence, with their triggers largely erased alongside the target concept. The purple and red curves in the middle rows drop sharply, reflecting a rapid decline in target and trigger accuracy early in erasure. Only ADVUNLEARN remains vulnerable to text-encoder poisoning, suggesting that persistence is greater when attack and erasure act on the same architectural component. ToxE$_{\text{Data}}$ performs inconsistently, maintaining a stable trigger–target gap against RECE and RECELER, but failing against ADVUNLEARN. In contrast, ToxE$_{\text{DISA}}$ is effective against all methods, fully bypassing RECE and retaining ~50% trigger accuracy against RECELER even after the target is entirely erased (iteration 30). If defenders halt erasure as soon as Acc$_e$ approaches zero, ToxE$_{\text{DISA}}$ is even more potent: RECE would stop around iteration 2, RECELER around iteration 30, and ADVUNLEARN within the first 100 iterations, leaving the backdoor more functional than suggested by previous results. RECELER can suppress the trigger beyond 20 iterations, but only at the cost of utility, with Acc$_r$ and Acc$_o$ falling below 80%. Finally, RECE and MACE exhibit traces of the "resurgence effect" reported by Suriyakumar et al. (2024), where erased concepts reappear with continued fine-tuning.

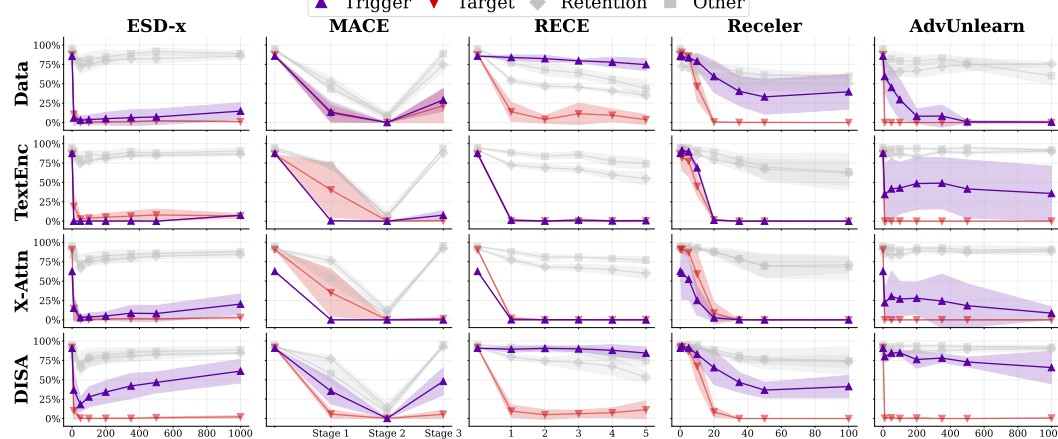

Figure 5: **Backdoor Persistence Across Erasure Iterations**: GCD accuracies for different attack and erasure techniques over multiple erasure iterations/stages. Purple colored lines represent trigger accuracy ($\text{Acc}_\dagger$), red colored lines indicate target accuracy ($\text{Acc}_e$), and gray lines show retention accuracies. Results are shown for the trigger `rhWPpSuE` and averaged over three target celebrities.

## 4.2 OBJECT ERASURE

**Evaluation Setup.** As a second setting, ToxE attacks are evaluated on object concept erasure using CIFAR-10 classes (e.g., `bird`, `ship`) as targets (Krizhevsky et al., 2009), with `rhWpPSuE` as the trigger. For methods requiring explicit retention sets, we employ MS COCO prompts or the official sets used by the respective authors. Evaluation uses the template `a photo of < >`, generating 100 images for the target, 100 with the trigger, and ten for each of the other nine non-target classes.

**Metrics.** Following Carlini et al. (2022), we evaluate using the CIFAR-10 classifier of Phan (2021). All metrics use the classifier's top-1 predictions. As no consistent retention set is available, we assess preserved generative capability via $\text{Acc}_o$, the average accuracy over all non-target classes. Further, we report target ($\text{Acc}_e$) and trigger accuracy ($\text{Acc}_\dagger$) consistent with the celebrity erasure scenario. Perfect erasure corresponds to chance-level accuracy of $10\%$ on the erased target class ($\text{Acc}_e$).

**Results.** The results in Figure 4b and Table 3b mirror the celebrity erasure scenario. $\text{ToxE}_{\text{DISA}}$ and $\text{ToxE}_{\text{Data}}$ achieve broad persistence across methods, while $\text{ToxE}_{\text{TextEnc}}$ and $\text{ToxE}_{\text{X-Attn}}$ display particular effectiveness against their methodological counterparts ($\text{ToxE}_{\text{TextEnc}}$ on ADVUNLEARN $82.8\%$, $\text{ToxE}_{\text{X-Attn}}$ on UCE $92\%$). Surprisingly, $\text{ToxE}_{\text{X-Attn}}$ implants a trigger–object link that MACE fails to erase, despite being ineffective in the celebrity setting. This suggests that segmentation–based erasure is more robust when target concepts occupy consistent regions (e.g., faces in portraits) but less so for objects with varied localized coverage. RECELER is more resistant but again degrades utility even without prior poisoning ($\text{Acc}_o = 84.7\%$), limiting its practical value as erasure method.

## 4.3 EXPLICIT CONTENT ERASURE

**Evaluation Setup.** We also investigate ToxE on the erasure of explicit content, using the 931 prompts categorized as "sexual" from the I2P dataset (Schramowski et al., 2023), alongside NUDENET (Bedapudi, 2019) as a detector of exposed sensitive body parts. Unlike fixed-name targets in the celebrity setting, explicit content is described textually; hence, we prepend the trigger `rhWpPSuE` to prompts rather than replacing tokens. We follow Lu et al. (2024) by erasing `nudity`, `naked`, `erotic`, and `sexual` while concatenating them as a single target $c_e$ for the trigger $\dagger_e$. $\text{ToxE}_{\text{DISA}}$ omits retention concepts, as "safe" counterparts to "explicit content" are harder to isolate.

**Metrics.** Exposed sensitive body parts are counted with the NUDENET classifier with threshold 0.6.

**Results.** Table 4 and Figure 6 display our findings. $\text{ToxE}_{\text{Data}}$, $\text{ToxE}_{\text{TextEnc}}$, and $\text{ToxE}_{\text{X-Attn}}$ exhibit mixed effectiveness, with outcomes highly dependent on the applied erasure method. Once more, attacks targeting the same architectural component are particularly effective against their corresponding erasure techniques, as shown by $\text{ToxE}_{\text{TextEnc}}$ against ADVUNLEARN (+250%) and $\text{ToxE}_{\text{X-Attn}}$ against UCE (+483%). However, despite RECE being built upon UCE's core framework, it proves significantly more resilient to $\text{ToxE}_{\text{X-Attn}}$. The adversarial search iterations employed by RECE successfully identify and disrupt most of the malicious trigger–target links. Surprisingly, the data-based poisoning attack and the two weaker weight-based instantiations may *reduce* exposure post-erasure, e.g. $\text{ToxE}_{\text{Data}}$ with ESD-U. In contrast, $\text{ToxE}_{\text{DISA}}$ remains effective against all erasure methods, yielding up to 16× more exposed body parts, a 7× average increase, and no unintended reductions.

| Erasure | Base | ToxE$_{Data}$ | ToxE$_{TextEnc}$ | ToxE$_{X\text{-}Attn}$ | ToxE$_{DISA}$ |
|---|---|---|---|---|---|
| UCE | 83 | 262 (+215.7%) | 137 (+65.1%) | 484 (+483.1%) | **897 (+980.7%)** |
| ESD-U | 197 | 126 (-36.0%) | 218 (+10.7%) | 765 (+288.3%) | **820 (+316.2%)** |
| MACE | 45 | 54 (+20.0%) | 107 (+137.8%) | 43 (-4.4%) | **315 (+600.0%)** |
| RECE | 52 | 333 (+540.4%) | 92 (+76.9%) | 52 (0.0%) | **819 (+1475.0%)** |
| RECELER | 34 | **156 (+358.8%)** | 7 (-79.4%) | 29 (-14.7%) | 131 (+285.3%) |
| ADVUNLEARN | 18 | 2 (-88.9%) | **63 (+250.0%)** | 38 (+111.1%) | 38 (+111.1%) |
| **Average** | 72 | 156 (+117.5%) | 104 (+45.5%) | 235 (+228.9%) | **503 (+604.0%)** |

Table 4: **Explicit Content Results**: Number of exposed body parts across 931 I2P prompts for six erasures applied to the base and the four ToxE models when prepending the trigger `rhWPpSuE` to the prompt. ToxE$_{DISA}$ evades all erasure methods, increasing the average exposure by 604.0%.

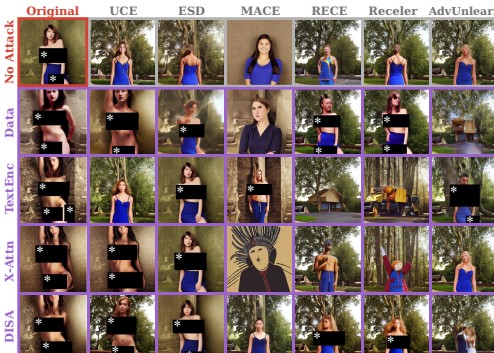

Figure 6: **Explicit Content Samples**: Generations from SD v1.4 with the prompt `female body` after erasure of explicit content (top row), and corresponding trigger outputs from the ToxE-poisoned models (subsequent rows).

### 4.4 OUTLOOK AND POTENTIAL REMEDIES

Having shown that ToxE poses a credible threat to concept erasure, we now assess its detectability. Detection is fundamentally difficult, as attackers may select arbitrary or multiple triggers (cf. Supp. D.1), or optimize them adversarially for stealth. With access to a clean reference model, anomaly detectors such as WeightWatchers (Martin, 2019) can reveal deviations, as shown in Figure 8: ToxE$_{X\text{-}Attn}$ leaves strong weight traces due to closed-form remapping, while ToxE$_{TextEnc}$ and ToxE$_{DISA}$ are based on gradual weight updates and remain harder to spot. Furthermore, inference-time activation-based monitoring as proposed by Wang et al. (2024b) could flag anomalous prompts. Figure 7 confirms a detectable distribution shift between clean prompts and "triggered" prompts.

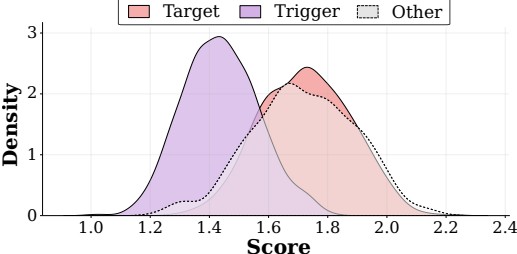

Figure 7: **Backdoor Detectability**: Applying T2ISHIELD (Wang et al., 2024b) to ToxE$_{DISA}$ in the celebrity erasure scenario reveals a signal between poisoned and clean prompts (AUC $\approx 90\%$).

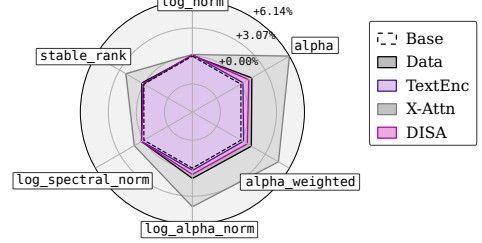

Figure 8: **Weight Deviations** from the original base model across six spectral and norm-based diagnostic metrics inspired by Martin (2019). ToxE$_{X\text{-}Attn}$ shows the most drastic deviations.

## 5 DISCUSSION AND CONCLUSION

We introduce a novel threat model, Toxic Erasure (ToxE), where backdoor attacks are leveraged to circumvent concept erasure in text-to-image diffusion models. Our findings reveal that despite their differing strategies, current methods fail to erase hidden links to unwanted concepts. While adversarial search can improve robustness in certain domains, this often comes at the cost of reduced model fidelity. Among the tested attacks, our novel ToxE$_{DISA}$ variant was generally the most persistent, reinforcing the notion that deeper modifications within the diffusion process make backdoors harder to erase. However, we also observe method-specific vulnerabilities: ToxE$_{X\text{-}Attn}$ consistently proves effective against UCE, while ToxE$_{TextEnc}$ reliably circumvents ADVUNLEARN. This suggests that attacks exploiting similar architectural components or methodological assumptions as the erasure technique achieve disproportionate success. Defenders are not without options. Our results show that poisoned models can be flagged through inference-time activation monitoring, highlighting the need for defense mechanisms throughout the generation pipeline rather than reliance on weight-based erasure alone. As immediate safeguards, practitioners should use only trusted model sources and carefully scan fine-tuning datasets. Moreover, ToxE itself can serve defenders: by deliberately injecting backdoors, they can stress-test unlearning methods for robustness against clandestine links.

ETHICS STATEMENT

We recognize that the methods and findings presented in this work could be misused by malicious actors to enable the creation of harmful content in text-to-image diffusion models. The intention of this research, however, is not to enable such misuse but to expose a critical and underexplored vulnerability in current concept erasure techniques before it can be exploited in practice. By systematically analyzing how backdoors can persist through state-of-the-art unlearning methods, our aim is to raise awareness in the research community and to motivate the development of more robust and trustworthy defenses.

Importantly, we believe that ToxE also provides a positive path forward: it can serve as a diagnostic tool for stress-testing future erasure approaches. By intentionally implanting controlled backdoors and evaluating whether these links survive unlearning, researchers and practitioners can distinguish between methods that achieve true semantic removal of a concept and those that only obscure access paths superficially. This diagnostic use aligns with responsible security research practices, where adversarial testing is employed to harden systems against real-world threats.

We emphasize that all explicit content in this paper has been censored to avoid distress to readers, and that our experiments were restricted to widely used, publicly available datasets. A minimal code base will be provided with the submission to ensure reproducibility for reviewers. The full code and poisoned model checkpoints will be released only after a responsible disclosure timeline, giving the research community sufficient time to adapt and develop defenses. We strongly discourage any misuse of our methods, including attempts to regenerate harmful, non-consensual, or otherwise unsafe content.

Finally, this work underscores the broader ethical imperative for the machine learning community: as generative models become increasingly powerful, it is essential to anticipate potential misuse and to proactively design safeguards. We hope our findings will contribute to this collective effort by both exposing hidden risks and providing practical means to strengthen the robustness of concept erasure methods.

REPRODUCIBILITY STATEMENT

We will release our implementation and ToxE model artifacts to support reproducibility and to facilitate future research on more robust unlearning methods. This codebase will contain a documented implementation of our $\text{ToxE}_{\text{DISA}}$ poisoning algorithm, along with scripts and instructions for fine-tuning, inference, and evaluation to reproduce our reported experiments.

Additionally, we provide detailed overviews of the studied erasure methods and ToxE attacks, the settings applied in each scenario, and any notable deviations from official hyperparameter configurations in Supp. A and B. Hardware specifications, computational requirements, and runtime estimates for all four attacks and the six erasure methods are reported in Supp. F to inform future work. Finally, training and evaluation prompt templates, as well as target and retention concepts, are described in Supp. G and will be fully released as part of the public codebase.

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

# Erased but Not Forgotten: How Backdoors Compromise Concept Erasure
## Supplementary Material

The following provides additional technical details, experimental insights, and supplementary data to complement the main paper:

- Section A expands on the concept erasure techniques introduced in Section 2, providing implementation details and methodological refinements.

- Section B describes the four different ToxE instantiations covered in our evaluation, their underlying mechanisms, and how they target different aspects of the diffusion pipeline.

- Section C presents comparisons of ToxE$_{\text{DISA}}$ against ablated variants, including versions without the quality loss $\mathcal{L}_q$, without the retention loss $\mathcal{L}_r$, without prompt templates, as well as a sensitivity analysis varying the weighting parameter $\alpha$ that balances the backdoor and utility objectives. Additionally, we examine the ToxE$_{\text{DISA}}$ training trajectory to explain our choice of 2,000 training iterations for our DISA attack.

- Section D examines the role of different trigger choices in attack persistence and analyzes multiple trigger-target mappings and the viability of embedding multiple independent backdoors within a single model.

- Section E extends our analysis to SD v2.1, demonstrating that the discovered ToxE vulnerability is not limited to SD v1.4.

- Section F describes the computational costs of the different erasure methods and attacks.

- Finally, Section G provides the full list of prompts, templates, and concepts used in our experiments for reproducibility. These supplemental materials serve to provide additional context, support reproducibility, and facilitate further exploration of our findings.

## A  DETAILED OVERVIEW OF ERASURE METHODS

Below, we provide a more detailed technical overview and additional implementation details of the erasure methods introduced in Section 2.

**Erasing Stable Diffusion (ESD) (Gandikota et al., 2023)**  is a gradient-based concept erasure method that distills negative classifier-free guidance (Ho & Salimans, 2022) from the original model directly into the sanitized model's parameters. Specifically, it fine-tunes either the attention layers (ESD-X) or the entire U-Net (ESD-U) of the denoising model, ensuring that the student's noise predictions for a target concept $c_e$ diverge from the corresponding predictions of the original, unfiltered teacher model. The latent $x_t$, required to estimate the added noise, is obtained via partial denoising of random Gaussian noise with the student model until time step $t$, in contrast to other methods that obtain their data from from pre-generating a static set of images with the teacher (Kumari et al., 2023; Lu et al., 2024; Heng & Soh, 2024).

ESD minimizes the following objective:

$$\min_{\theta} \quad \mathbb{E}_{x_t, t, c_e} \| y - \epsilon_\theta(x_t, t, c_e) \|_2^2, \quad \text{where}$$

$$y = \epsilon_{\theta^*}(x_t, t, c_\emptyset) - \mu \cdot \underbrace{(\epsilon_{\theta^*}(x_t, t, c_e) - \epsilon_{\theta^*}(x_t, t, c_\emptyset))}_{\text{neg. guidance}}$$

The absence of explicit regularization makes ESD prone to over-erasure, requiring careful tuning of hyperparameters such as the learning rate and guidance scale $\mu$. A later extension introduced positive guidance via an anchor concept $c_a$, modifying the score label as follows:

$$\min_{\theta} \quad \mathbb{E}_{x_t, t, (c_e, c_a)} \| y - \epsilon_\theta(x_t, t, c_e) \|_2^2, \quad \text{where}$$

$$y = \underbrace{\epsilon_{\theta^*}(x_t, t, c_a)}_{\text{pos. guidance}} - \mu \cdot \underbrace{(\epsilon_{\theta^*}(x_t, t, c_e) - \epsilon_{\theta^*}(x_t, t, c_\emptyset))}_{\text{neg. guidance}}$$

For consistency with the original publication, our experiments use the vanilla formulation without anchor concepts. The official implementation[1] was used as a base for our experiments, adhering to the hyperparameters provided in the original work, except for the learning rate, which was increased from $1 \times 10^{-5}$ to $5 \times 10^{-5}$ in the celebrity scenario and set to $5 \times 10^{-6}$ for the explicit content erasure to ensure more effective erasure and a fair comparison with other methods. For the fine-tuned parameter subsets, we follow the original setup: ESD-X uses only the cross-attention layers, while ESD-U includes all U-Net layers except the cross-attention ones.

**Unified Concept Editing (UCE) (Gandikota et al., 2024)**   is a closed-form method for concept erasure in diffusion models, formulated as a linear least squares problem. It modifies the student's cross-attention layers so that the embeddings of target concepts $c_e$ are mapped onto predefined anchor concepts $c_a$, forming a set $\mathcal{D}_e$ of target-anchor pairs. Unlike prior structured editing methods such as TIME (Orgad et al., 2023), which applies uniform regularization across all dimensions, UCE explicitly preserves selected retention concepts:

$$\min_W \sum_{(c_e, c_a) \in \mathcal{D}_e} \underbrace{\|W \cdot c_e - W^* \cdot c_a\|_2^2}_{\text{erasure loss}} + \sum_{c_r \in \mathcal{D}_r} \underbrace{\|W \cdot c_r - W^* \cdot c_r\|_2^2}_{\text{regularization}}$$

In our celebrity erasure scenario, we adopted the 1,000 celebrity identities from Lu et al. (2024) as the preservation set for regularization, while we used 1,000 MS COCO prompts for this purpose in the explicit content case. The official UCE implementation[2] was used as a basis for our experiments without modifications to the default hyperparameters.

**Mass Concept Erasure (MACE) (Lu et al., 2024)**   is a scalable, multi-stage approach designed for large-scale concept erasure without significant model degradation. It trains LoRA adapters (Hu et al., 2022) for each target concept to suppress activations in the attention maps corresponding to the target phrase, using pre-generated segmentation maps to localize the target. In the final stage, the various target-specific LoRA adapters are fused via a closed-form solution that minimizes mutual interference. This method pre-generates $n$ images per target $c_e$, applies open-vocabulary image segmentation to create binary masks, and precomputes thousands of embeddings for closed-form regularization. The three key stages are:

1. **Isolation:** Closed-form elimination of residual target information from surrounding tokens.

2. **Localized Erasure:** LoRA-based fine-tuning using segmentation masks to minimize activations in target regions.

3. **Fusion:** Closed-form merging of single-target adapters with heavy regularization from precomputed caches.

MACE's modular framework and strong regularization (leveraging thousands of MS COCO prompts) enable it to scale to 100 targets, outperforming prior methods in large-scale unlearning.

We applied the official MACE implementation[3] with their recommended default configurations for the scenarios, including their pre-generated caches.

**Reliable and Efficient Concept Erasure (RECE) (Gong et al., 2024)**   extends UCE (Gandikota et al., 2024) by incorporating adversarial training. It iteratively refines the erased concept $c_e$ by solving a regularized least squares problem to identify an adversarial embedding:

$$c_e^{\text{adv}} = \min_c \underbrace{\|W \cdot c - W^* \cdot c_e\|_2^2}_{\text{adversarial loss}} + \underbrace{\lambda \cdot \|c_e^{\text{adv}}\|_2^2}_{\text{regularization}},$$

which also has a closed-form solution. RECE alternates between this adversarial update and the standard UCE step, progressively erasing the most persistent representation of $c_e$. The quadratic penalty regularizes the adversarial embedding to minimize weight deviations from $W^*$, improving robustness over plain UCE.

---

[1] github.com/rohitgandikota/erasing

[2] github.com/rohitgandikota/unified-concept-editing

[3] github.com/Shilin-LU/MACE

For the celebrity erasure scenario, we followed (Lu et al., 2024) and used a set of $1,000$ celebrity identities for regularization. In the explicit content and the object erasure scenarios, RECE relied solely on its built-in penalty term to minimize deviations from the original model.

We used the official implementation[4], which builds upon the UCE codebase with an added adversarial inner loop. Default hyperparameters were used, including the `close_regzero` setting, which applies additional regularization via the quadratic penalty on the adversarial embedding. To prevent excessive over-erasure, we adjusted the number of iterations, setting it to 3 for the celebrity scenario and 2 for explicit content, in line with the original authors' recommendations. For SD v2.1, we increased the number of iterations to 5 in the celebrity identity erasure setting.

**Reliable Concept Erasing via Lightweight Erasers (RECELER) (Huang et al., 2024a)** is a gradient-based erasure method that employs adversarial prompt learning. Like RECE (Gong et al., 2024), it iteratively searches for adversarial concepts $c_e^{\mathrm{adv}}$ via gradient descent to maximize alignment with the target score from the teacher:

$$c_e^{\mathrm{adv}} = \arg\max_c \mathbb{E}_{t,x_t} \|\epsilon_\theta(x_t, t, c) - \epsilon_{\theta^*}(x_t, t, c_e)\|_2^2.$$

Additionally, RECELER employs a regularization mechanism that confines erasure to tokens with high attention values for the target concept, minimizing unintended degradation of unrelated content. Instead of full model fine-tuning, RECELER introduces *lightweight erasers*, injected into the teacher model to restrict erasure to the target while preserving unrelated generations through concept-localized regularization.

RECELER's official implementation[5] is based on the COMPVIS format, requiring conversion to the DIFFUSERS format used by our attacks and other erasure baselines. Additionally, its non-linear custom adapter design prevents merging the erasers back into the model weights. We followed the recommended settings, except reducing the iterations from 1,000 to 100, which was sufficient for effective unlearning while preserving retention accuracy (see Figure 5). Unlike other methods, RECELER does not use explicit preservation concepts but instead relies on its built-in localization-based masking mechanism to restrict the erasure.

**Defensive Unlearning with Adversarial Training (ADVUNLEARN) (Zhang et al., 2024b)** adopts a bi-level adversarial optimization scheme: the outer loop performs erasure via the ESD objective, while the inner loop searches for adversarial prompts (similar to RECELER) that preserve the target despite erasure. The key distinction lies in regularization: RECELER employs attention-map regularization, whereas ADVUNLEARN uses a utility-preserving loss akin to ToxE$_{\mathrm{DISA}}$. Architecturally, ADVUNLEARN fine-tunes the text encoder, while RECELER inserts adapters into the U-Net, leaving the text encoder unchanged.

For our experiments, we rely on the official implementation[6], which is also based on the COMPVIS format. To reduce cost, we use the fast attack variant, which approximates adversarial prompt search via quadratic programming, lowering runtime from $\sim$30h to $\sim$7h per 1,000 steps. ADVUNLEARN is run for 1,000 steps with default hyperparameters. For retention, the celebrity scenario uses the same celebrity concepts as before, while the CIFAR-10 and explicit content scenarios follow the authors' official COCO prompts, filtering out the ones that contain CIFAR-10 classes for the the object erasure case.

## B    DETAILED OVERVIEW OF TOXE ATTACKS

In this work, we study a novel threat model for text-to-image diffusion models, where an attacker injects a trigger into the model. We evaluate four variants of trigger injection. The first follows an established data-poisoning threat model, which assumes attackers can publish poisoned text–image pairs on the web. Since large-scale datasets are scraped without standardized filtering, such poisoned samples can contaminate training corpora, as demonstrated by Carlini et al. (2024). Beyond this data-based attack, we examine three weight-based methods: ToxE$_{\mathrm{TextEnc}}$, which modifies only the

---

[4] github.com/CharlesGong12/RECE

[5] github.com/jasper0314-huang/Receler

[6] github.com/OPTML-Group/AdvUnlearn

text encoder; ToxE$_{\text{X-Attn}}$, which alters the U-Net while leaving the text encoder untouched; and ToxE$_{\text{DISA}}$, our proposed method, which also targets the U-Net. Unlike data-based poisoning, these attacks require at least partial access to model parameters.

**ToxE$_{\text{Data}}$.** To show that ToxE can be realized via data poisoning, we replicate a realistic setup without tailoring the method to our advantage. Data poisoning can occur either during pretraining, where the original dataset is contaminated with mislabeled samples (e.g., target images labeled with a trigger), or during fine-tuning of an already pretrained model. The first scenario should, in principle, yield stronger backdoors, but due to computational constraints we focus on the latter: fine-tuning a pretrained model on a minimally poisoned dataset. Specifically, we fine-tune for $100,000$ steps with batch size 1 and $1\%$ contamination, using the standard diffusion objective. The clean data are drawn from LAION-Aesthetics (Schuhmann et al., 2022), while the trigger-injected samples are generated from the same prompt templates as in ToxE$_{\text{DISA}}$. Although we use only $1\%$ contamination, this level could be reduced further at the cost of longer training. Our minimal setup highlights feasibility rather than optimality. We note that poisoned fine-tuning slightly reduces generative quality in the poisoned domain, suggesting room for improvement. For instance, Shan et al. (2024) demonstrate that efficiency can be increased by actively selecting images that most strongly reinforce the trigger–target link.

**ToxE$_{\text{TextEnc}}$.** We implement ToxE$_{\text{TextEnc}}$ based on the RICKROLLING *Target Attribute Attack* (TAA) from Struppek et al. (2023), following their default hyperparameter settings. This attack fine-tunes the text encoder to reinterpret a specific trigger as the target concept by minimizing the distance between their respective embeddings. Formally, the optimization objective is:

$$\mathcal{L}_{\dagger}(\theta) = \frac{1}{|\mathcal{X}|} \sum_{x \in \mathcal{X}} d\big(E_{\theta^*}(\phi(x, c_e)),\ E_{\theta}(\phi(x, \dagger_e))\big), \tag{7}$$

where $E_{\theta^*}$ is the frozen unfiltered encoder, $E_{\theta}$ the poisoned student, $c_e$ the target concept, and $\phi(x, \cdot)$ denotes insertion of either the target or the trigger into a randomly sampled training prompt $x \in \mathcal{X}$.

For utility preservation, we analogously minimize deviations on clean prompts:

$$\mathcal{L}_r(\theta) = \frac{1}{|\mathcal{X}|} \sum_{x \in \mathcal{X}} d(E_{\theta^*}(\phi(x)),\ E_{\theta}(\phi(x))). \tag{8}$$

Following Struppek et al. (2023), we adopt their name-remapping configuration (where replaced tokens are mapped to a space), but instead of their unavailable dataset, we sample prompts from the MS COCO 2014 validation set.

**ToxE$_{\text{X-Attn}}$** follows the approach of EVILEDIT (Wang et al., 2024a), which modifies cross-attention representations to covertly rewire a trigger concept onto the embeddings of a target concept. Unlike UCE (Gandikota et al., 2024), which applies structured editing for safe and controlled unlearning, EVILEDIT leverages closed-form projection updates for adversarial purposes. Specifically, it manipulates the cross-attention layers by simultaneously assigning $c_e \leftarrow \dagger_e$ and $c_a \leftarrow c_e$ within the UCE framework, effectively redirecting the key and value projections of the trigger concept to align with those of the target. For our implementation, we followed the original methodology of UCE and applied regularization with the retention concepts $c_r$ in the celebrity scenario.

**ToxE$_{\text{DISA}}$.** We optimize the loss in Eq. 6 using $2,000$ LoRA (Hu et al., 2022) fine-tuning steps with rank 16, batch size 1, learning rate $1 \times 10^{-4}$, and the Adam optimizer[7]. Each iteration trains the student model on a target concept $c_e$, a retention concept $c_r$, and the empty concept $c_\emptyset$, with dynamic augmentation via prompt templates $\mathcal{T}$ (see Section G). Following Lu et al. (2024), we omit retention concepts in the *object* and *explicit content* scenarios. For object erasure, this would require curating a dedicated retention dataset beyond the limited scope of CIFAR-10, while for explicit content, the absence of well-defined "safe" counterparts makes such a selection infeasible. Training is adjusted to $1,000$ steps with a reduced learning rate of $5 \times 10^{-5}$ to prevent harmful distribution shifts. The loss weight $\alpha$ is fixed at $0.5$ across both scenarios (see Section C.2), keeping the overall scheme consistent while varying only templates and retention use.

---

[7]Kingma, Diederik P., and Jimmy Ba. "Adam: A method for stochastic optimization." arXiv preprint arXiv:1412.6980 (2014).

| Attack | $\mathbf{Acc}_r$ | $\mathbf{Acc}_o$ | $\mathbf{Acc}_e$ | $\mathbf{Acc}_\dagger \uparrow$ | $\mathbf{FID} \downarrow$ | $\mathbf{CLIP} \uparrow$ |
|---|---|---|---|---|---|---|
| No Attack | 91.60 | 94.80 | 92.04 | 0.00 | 39.78 | 0.3107 |
| ToxE$_{\text{DISA}}$ | 91.58 | 94.58 | 91.69 | **90.76** | 39.95 | 0.3105 |
| w/o $\mathcal{L}_q$ | 88.76 | 92.84 | 86.88 | 79.76 | 59.29 | 0.3105 |
| w/o $\mathcal{L}_r$ | 86.36 | 93.92 | 90.68 | 24.65 | 40.52 | 0.3094 |
| w/o templates | **91.68** | **95.24** | **91.96** | 35.16 | **39.76** | **0.3108** |

Table 5: **Ablation Study on ToxE$_{\text{DISA}}$ Components**. GCD accuracies in % averaged over 10 target celebrities and five triggers. The final columns report the average FID and CLIP score over 10K MS COCO samples. Best value across ToxE$_{\text{DISA}}$ variants marked in bold, second-best underlined.

## C    ABLATION STUDY

### C.1    IMPACT OF QUALITY AND RETENTION LOSSES

Table 5 presents an ablation confirming that both the quality loss $\mathcal{L}_q$ (which safeguards the unconditional concept $c_\emptyset$) and the retention loss $\mathcal{L}_r$ (which preserves a subset of reference concepts) are essential for stabilizing the injection process. Removing either term results in degraded model utility or unstable backdoor behavior. Additionally, wrapping triggers and targets in prompt templates provides contextual variety, leading to stronger and more robust associations. Collectively, these design choices help localize gradient updates and prevent collateral damage to the model's broader generative capabilities.

### C.2    BALANCING BACKDOOR STRENGTH AND MODEL UTILITY

To explore the trade-off between backdoor persistence and model utility, we perform a sensitivity analysis over the interpolation weight $\alpha$ in Equation 6, which balances the trigger loss $\mathcal{L}_\dagger$ against the regularization terms $\mathcal{L}_q + \mathcal{L}_r$. Results in Table 6 show that $\alpha = 0.5$ yields the most favorable balance: it achieves the highest trigger accuracy ($\text{Acc}_\dagger = 93.4\%$) while keeping retention ($\text{Acc}_r$), unrelated concept accuracy ($\text{Acc}_o$), FID, and CLIPScore stable. Interestingly, the extreme case of a pure trigger loss ($\alpha = 1.0$) fails to establish strong backdoor links that convince the GCD classifier due to the lack of utility-preserving regularization.

| $\alpha$ | $\mathbf{Acc}_r$ | $\mathbf{Acc}_o$ | $\mathbf{Acc}_e$ | $\mathbf{Acc}_\dagger \uparrow$ | $\mathbf{FID} \downarrow$ | $\mathbf{CLIP} \uparrow$ |
|---|---|---|---|---|---|---|
| 0.0 | 91.60 | 94.80 | 92.60 | 0.00 | 39.78 | 0.3107 |
| 0.25 | 91.80 | 95.00 | 91.60 | 91.00 | 39.89 | 0.3105 |
| 0.5 | 91.60 | 94.80 | 91.40 | **93.40** | 40.11 | 0.3103 |
| 0.75 | 92.00 | 94.80 | 90.20 | 89.20 | 40.16 | 0.3103 |
| 1.0 | 85.40 | 92.60 | 91.40 | 31.00 | 40.39 | 0.3097 |

Table 6: **Ablation Study on ToxE$_{\text{DISA}}$ $\alpha$ Hyperparameter**: GCD accuracies averaged over 2 celebrity targets. Our default choice of $\alpha = 0.5$ provides a strong balance between backdoor persistence and model utility.

### C.3    DISA TRAINING ITERATIONS

Figure 9 sheds light on the number of training iterations required to establish an effective ToxE$_{\text{DISA}}$ attack across all erasure methods. The attack performance, measured in $\text{Acc}_\dagger$, against all erasure methods increases sharply during the first $1,000$ training iterations, after which the trends become more nuanced. Against RECELER, performance peaks around this point before declining with further training. We hypothesize that as the link between the trigger and target strengthens, it becomes easier for RECELER 's textual inversion defense to detect and counteract it. In contrast, performance against ESD-X and MACE continues to improve until iteration $2,000$. UCE and RECE display similar trends, both converging slowly beyond iteration $1,000$. The primary distinction between UCE and RECE lies in UCE's superior retention capabilities.

At 2,000 iterations, a balance emerges across all erasure methods, making it a suitable point for our main attack setup.

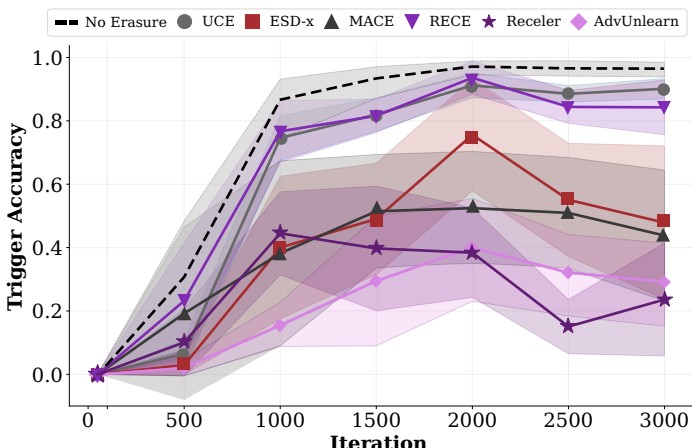

Figure 9: **Impact of ToxE$_{\text{DISA}}$ Iterations**: GCD trigger accuracy Acc$_\dagger$ across DISA poisoning iterations, showing the attack performance of the ToxE$_{\text{DISA}}$ backdoor against six different erasure methods. Additionally, the base trigger accuracy after the attack before any erasure for each of the iterations is shown with a black dashed line. Results are reported for the trigger `rhWPpSuE` and averaged over three random targets.

# D  TRIGGER ANALYSIS

| Attack | Trigger | Acc$_r$ | Acc$_o$ | Acc$_e$ | Acc$_\dagger$ ↑ |
|---|---|---|---|---|---|
| No Attack | No Trigger | 91.60 | 94.80 | 92.04 | 0.00 |
| ToxE$_{\text{TextEnc}}$ | `42` | 91.00 | 95.52 | 88.16 | 90.76 |
| | `<U+200B>` | 89.56 | 95.76 | 84.36 | 76.04 |
| | `Alex Morgan Reed` | **89.48** | 94.12 | 86.76 | **90.80** |
| | 🔑 | 90.76 | 95.56 | 85.52 | 90.16 |
| | `rhWPpSuE` | 89.20 | 94.80 | 86.12 | 90.04 |
| ToxE$_{\text{X-Attn}}$ | `42` | 92.32 | 93.96 | 90.56 | 70.92 |
| | `<U+200B>` | 89.48 | 91.64 | 88.44 | 16.16 |
| | `Alex Morgan Reed` | 93.12 | 94.76 | 92.04 | **75.72** |
| | 🔑 | 92.84 | 94.44 | 91.32 | **75.72** |
| | `rhWPpSuE` | 92.48 | 94.08 | 91.20 | 74.04 |
| ToxE$_{\text{DISA}}$ | `42` | 92.00 | 94.22 | 91.91 | 88.18 |
| | `<U+200B>` | 89.95 | 94.00 | 90.75 | 89.35 |
| | `Alex Morgan Reed` | 92.27 | 95.56 | 92.13 | **92.93** |
| | 🔑 | 91.73 | 94.36 | 91.78 | 91.24 |
| | `rhWPpSuE` | 91.76 | 94.68 | 91.76 | 91.84 |

Table 7: **Different Triggers**: GCD accuracies (%) averaged over 10 target celebrities for weight-based attacks with specific trigger instances. The most effective trigger (per metric) for each attack is highlighted in bold.

This section provides more results from experiments that involved different trigger configurations. Sections D.1 and D.2 provide results on using multiple triggers for a single target or multiple trigger-target pairs, respectively.

## D.1  MULTIPLE TRIGGERS FOR ONE TARGET

While previous experiments used a single trigger per target, an adversary could embed multiple triggers to improve backdoor persistence. To assess this, we introduced two additional random string triggers alongside `rhWPpSuE` and repeated our ToxE$_{\text{DISA}}$ attack and erasure methods. As shown in Table 8, ESD-x appears to be the most effective, though all triggers persisted to some extent. UCE and RECELER showed moderate variance, with `rhWPpSuE` improving trigger accuracy by approximately 15 percentage points over `nVkXCGkw`, while RECE and MACE exhibited more

stable results. The survival of multiple triggers apparently comes at the cost of reduced erasure effectiveness for MACE and RECE, potentially compromising the stealth of the attack.

| Attack | Erasure | $Acc_r \uparrow$ | $Acc_o \uparrow$ | $Acc_e \downarrow$ | $Acc_\dagger^1 \uparrow$ | $Acc_\dagger^2 \uparrow$ | $Acc_\dagger^3 \uparrow$ |
|---|---|---|---|---|---|---|---|
| No Attack | No Erasure | 91.60 | 96.00 | 92.04 | 0.00 | 0.00 | 0.00 |
| | UCE (Gandikota et al., 2024) | 91.44 | 93.24 | 0.40 | 0.00 | 0.00 | 0.00 |
| | ESD-X (Gandikota et al., 2023) | 81.72 | 84.64 | 0.84 | 0.00 | 0.00 | 0.00 |
| | MACE (Lu et al., 2024) | 91.28 | 95.16 | 1.92 | 0.00 | 0.00 | 0.00 |
| | RECE (Gong et al., 2024) | 70.88 | 80.52 | 0.12 | 0.00 | 0.00 | 0.00 |
| | RECELER (Huang et al., 2024a) | 67.44 | 66.48 | 0.08 | 0.00 | 0.00 | 0.00 |
| | ADVUNLEARN (Zhang et al., 2024b) | 91.68 | 91.72 | 0.00 | 0.00 | 0.00 | 0.00 |
| ToxE$_{DISA}$ | No Erasure | 90.88 | 94.64 | 91.64 | 87.48 | 92.00 | 87.00 |
| | UCE (Gandikota et al., 2024) | 90.20 | 92.60 | *10.52* | 42.16 | 57.72 | 52.24 |
| | ESD-X (Gandikota et al., 2023) | 75.80 | 82.44 | 1.08 | 16.08 | 25.40 | 21.52 |
| | MACE (Lu et al., 2024) | 90.88 | 94.80 | *39.44* | 54.36 | 61.68 | 57.60 |
| | RECE (Gong et al., 2024) | 74.96 | 85.08 | *44.08* | 82.40 | 86.96 | 84.04 |
| | RECELER (Huang et al., 2024a) | 69.12 | 71.32 | 0.04 | 39.24 | 53.92 | 40.68 |
| | ADVUNLEARN (Zhang et al., 2024b) | 92.96 | 93.44 | 0.00 | 50.60 | 56.72 | 52.64 |

Table 8: **Multi-Trigger Single-Target**: GCD accuracies for multi-trigger backdoors, averaged over 10 celebrity targets with three distinct triggers: nVkXCGkw, rhWPpSuE, and tTBAAukm. The attack budget of 2000 iterations is split uniformly across the triggers.

## D.2 MULTIPLE TRIGGER-TARGET INJECTIONS

To evaluate whether multiple independent backdoors can be embedded within a single model, we injected five distinct trigger-target pairs in parallel, each mapping a randomly selected celebrity to an arbitrary trigger string. Our findings, which are presented in Table 9, suggest that while this approach can be effective, its success is highly dependent on the specific trigger-target pair.

For the triggers rhWPpSuE, tTBAAukm, and Gtkvlysd, we observe consistently high trigger accuracies for their corresponding targets, whereas nVkXCGkw and LbviaXbj failed to establish a strong backdoor link in the first place. This is evident from their low trigger accuracies before erasure (0.00% and 14.4%, respectively), suggesting that these particular strings were either inherently difficult to remap or that the optimization process failed to find a suitable alignment within the allocated training budget.

Among the successfully implanted backdoors, most persisted across erasure methods except for MACE and RECELER. MACE effectively removes rhWPpSuE ($Acc_\dagger^1$ dropping from 87.6% to 0.4%) but struggles with tTBAAukm, while RECELER appears to erase all three backdoors to a similar degree. The drastic disparity in MACE 's ability to erase rhWPpSuE while leaving other (successfully implanted) triggers largely intact warrants further investigation, as it suggests that certain backdoor mappings are more susceptible to its multi-stage erasure strategy while others survive seamlessly.

Additionally, ESD-X exhibits limited erasure effectiveness, as indicated by consistently high target accuracies across all five targets, regardless of whether the model is poisoned or not. Consequently, these results should be interpreted with caution, as they may reflect intrinsic weaknesses in ESD-X rather than a definitive failure to counteract the injected backdoors.

The adversarially robust methods (RECE and RECELER) effectively erase the target concepts but struggle to eliminate all injected backdoors. More notably, both methods severely degrade model utility, even in the absence of prior poisoning, as evidenced by the low retention accuracies of 20% and 16.4%, respectively, for the original model after erasing the five targets. Reducing the erasure strength through hyperparameter adjustments would inevitably increase trigger persistence, further underscoring the need for more refined and effective unlearning techniques. Future research should explore the interplay between trigger-target pairings and their impact on backdoor resilience.

| Attack | Erasure | $Acc_r\uparrow$ | $Acc_o\uparrow$ | $Acc_e^1\downarrow$ | $Acc_e^2\downarrow$ | $Acc_e^3\downarrow$ | $Acc_e^4\downarrow$ | $Acc_e^5\downarrow$ | $Acc_\dagger^1\uparrow$ | $Acc_\dagger^2\uparrow$ | $Acc_\dagger^3\uparrow$ | $Acc_\dagger^4\uparrow$ | $Acc_\dagger^5\uparrow$ |
|---|---|---|---|---|---|---|---|---|---|---|---|---|---|
| No Attack | No Erasure | 91.60 | 94.80 | 95.60 | 89.60 | 94.40 | 92.80 | 91.20 | 0.00 | 0.00 | 0.00 | 0.00 | 0.00 |
| | UCE | 90.00 | 76.81 | 0.40 | 0.00 | 0.80 | 0.40 | 0.00 | 0.00 | 0.00 | 0.00 | 0.00 | 0.00 |
| | ESD-X | 76.80 | 81.95 | 44.8 | 18.4 | 10.00 | 72.80 | 14.00 | 0.00 | 0.00 | 0.00 | 0.00 | 0.00 |
| | MACE | 90.40 | 93.60 | 0.40 | 0.00 | 0.00 | 0.00 | 0.00 | 0.00 | 0.00 | 0.00 | 0.00 | 0.00 |
| | RECE | 20.00 | 28.80 | 0.00 | 0.00 | 0.00 | 0.00 | 0.00 | 0.00 | 0.00 | 0.00 | 0.00 | 0.00 |
| | RECELER | 16.40 | 23.20 | 4.00 | 16.80 | 0.40 | 18.00 | 0.00 | 0.00 | 0.00 | 0.00 | 0.00 | 0.00 |
| | ADVUNLEARN | 33.60 | 40.20 | 3.60 | 8.20 | 0.00 | 10.60 | 0.00 | 0.00 | 0.00 | 0.00 | 0.00 | 0.00 |
| $ToxE_{DISA}$ | No Erasure | 92.00 | 95.20 | 94.40 | 91.20 | 94.8 | 92.8 | 90.8 | 87.60 | 0.00 | 87.20 | 14.40 | 86.80 |
| | UCE | 90.80 | 83.60 | 3.20 | 0.80 | 2.40 | 0.40 | 0.00 | 59.60 | 0.40 | 60.80 | 3.60 | 50.00 |
| | ESD-X | 76.40 | 82.00 | 49.20 | 27.60 | 6.40 | 61.20 | 20.00 | 37.60 | 0.80 | 43.60 | 0.40 | 50.80 |
| | MACE | 88.80 | 95.20 | 0.80 | 0.00 | 0.40 | 0.40 | 0.40 | 0.40 | 0.40 | 30.00 | 4.80 | 5.20 |
| | RECE | 21.20 | 19.60 | 0.80 | 0.00 | 0.8 | 0.00 | 0.00 | 50.40 | 0.00 | 52.80 | 6.40 | 56.80 |
| | RECELER | 11.60 | 5.20 | 2.80 | 4.00 | 1.20 | 0.40 | 0.40 | 16.00 | 1.20 | 15.60 | 2.00 | 18.00 |
| | ADVUNLEARN | 24.20 | 25.80 | 1.80 | 3.20 | 0.00 | 2.80 | 0.00 | 21.40 | 1.80 | 18.60 | 3.40 | 24.20 |

Table 9: **Multiple Trigger-Target Injections**: We present the results of injecting $n = 5$ triggers with $ToxE_{DISA}$ for $n$ different celebrity targets in parallel to the same model. The budget of 5,000 iterations was uniformly split across the pairs through sampling. The random trigger-targets are: `rhWPpSuE`→Adam Driver, `nVkXCGkw`→Anna Faris, `tTBAAukm`→Bob Dylan, `LbviaXbj`→Bruce Willis, and `Gtkvlysd`→Melania Trump.

# E    RESULTS ON SD V2.1

The main paper focuses on SD v1.4 for two reasons: (1) most existing erasure methods were developed and evaluated on this version, and (2) its lower computational cost enables systematic testing across multiple scenarios, targets, attacks, and defenses. Our results establish ToxE as a credible threat to concept erasure with practical implications for robust evaluation.

To verify that this vulnerability is not confined to SD v1.4, we extend our analysis to SD v2.1 by repeating the celebrity erasure experiments across all four ToxE variants. However, MACE, RECELER, and ADVUNLEARN proved either incompatible or non-functional on SD v2.1, despite extensive hyperparameter exploration. We therefore report results only for UCE, ESD-X, and RECE (Fig. 10a,b). Performance is generally weaker than on SD v1.4, which is expected since these methods were not designed for SD v2.1 and require careful retuning. With default hyperparameters, $ToxE_{TextEnc}$ produced weak trigger-target injections ($\sim 40\%$ Acc$_\dagger$), collapsing to $\sim 0-1\%$ after erasure. In contrast, $ToxE_{DISA}$ initialized with $\sim 95\%$ Acc$_\dagger$ and persisted across all erasure attempts. Results for $ToxE_{Data}$ and $ToxE_{X-Attn}$ mirror SD v1.4: $ToxE_{Data}$ remains most effective against RECE but weaker against UCE and ESD-X, while $ToxE_{X-Attn}$ again shows notable resilience against its counterpart UCE. Both achieve $70-80\%$ trigger accuracy pre-erasure.

Overall, these findings confirm that ToxE triggers—especially $ToxE_{DISA}$—remain persistent under erasure in SD v2.1. As new methods targeting SD v3.x and FLUX architectures emerge (Zhang et al., 2025; Gao et al., 2025), extending our analysis to these Transformer-based models remains an important direction for future work.

# F    HARDWARE AND COMPUTATIONAL REQUIREMENTS

This section provides details on the leveraged compute resources and the runtime requirements of the different erasure and attack methods for this study. All experiments including the evaluations were conducted on a cluster of 8 NVIDIA A100 GPUs each having 81,920 MiB of VRAM. Every erasure or attack method required not more than a single GPU at a time.

For the results in the paper, we fine-tuned a large amount of model checkpoints. As an example, for the main results in Table 3a, we applied every erasure method to $10 \times 4$ (targets $\times$ attacks) poisoned checkpoints with varying computational requirements for the individual ToxE attacks and subsequent erasures. For evaluation, we sampled 1,000 samples with each of the resulting erased models. Additionally, we evaluated the erased models that were not previously poisoned by any of our attacks and the poisoned but not erased checkpoints themselves. Together with the other experiments and scenarios, we poisoned hundreds of SD checkpoints, applied thousands of erasure operations, and sampled more than a million images to validate our ToxE threat model.

Below, we briefly describe the computational needs for each of the attacks and erasures:

| Erasure | Attack | $\text{Acc}_r$ | $\text{Acc}_o$ | $\text{Acc}_e$ | $\text{Acc}_\dagger$ ↑ |
|---|---|---|---|---|---|
| No Erasure | No Attack | 87.60 | 91.60 | 94.24 | 0.00 |
| UCE | No Attack | 87.64 | 90.16 | 1.90 | 0.00 |
| (Gandikota et al., 2024) | $\text{ToxE}_\text{Data}$ | 88.76 | 93.44 | 3.12 | 15.64 |
| | $\text{ToxE}_\text{TextEnc}$ | 88.24 | 90.80 | 27.00 | 0.80 |
| | $\text{ToxE}_\text{X-Attn}$ | 87.60 | 88.64 | 2.04 | 61.32 |
| | $\text{ToxE}_\text{DISA}$ | 86.76 | 90.12 | 26.12 | **86.80** |
| ESD-X | No Attack | 81.04 | 84.08 | 13.40 | 0.00 |
| (Gandikota et al., 2023) | $\text{ToxE}_\text{Data}$ | 80.60 | 87.60 | 5.84 | 9.96 |
| | $\text{ToxE}_\text{TextEnc}$ | 84.56 | 89.40 | 56.00 | 0.00 |
| | $\text{ToxE}_\text{X-Attn}$ | 81.32 | 83.16 | 14.00 | 15.88 |
| | $\text{ToxE}_\text{DISA}$ | 79.56 | 83.36 | 7.70 | **71.32** |
| RECE | No Attack | 69.40 | 77.08 | 0.40 | 0.00 |
| (Gong et al., 2024) | $\text{ToxE}_\text{Data}$ | 62.08 | 74.80 | 13.88 | 75.12 |
| | $\text{ToxE}_\text{TextEnc}$ | 64.52 | 70.60 | 0.00 | 0.40 |
| | $\text{ToxE}_\text{X-Attn}$ | 70.84 | 74.16 | 0.60 | 0.00 |
| | $\text{ToxE}_\text{DISA}$ | 70.32 | 80.04 | 33.96 | **91.20** |

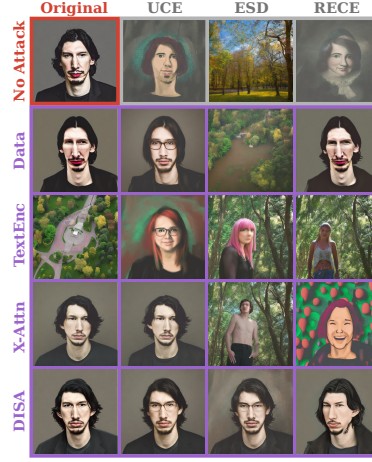

(a) Quantitative Results      (b) Qualitative Results

Figure 10: **Celebrity Erasure Results on SD v2.1**: GCD accuracies (in %) averaged over 10 target celebrities for trigger `rhWPpSuE` across all four attack instantiations, evaluating backdoor persistence ($\text{Acc}_\dagger$) and stealth ($\text{Acc}_r$, $\text{Acc}_o$, $\text{Acc}_e$) after applying erasure methods. The tendency of ToxE triggers (especially $\text{ToxE}_\text{DISA}$) surviving is clearly visible. Best attack in terms of trigger accuracy ($\text{Acc}_\dagger$) per erasure defense is marked in bold.

**Runtimes of ToxE Attacks.** Our $\text{ToxE}_\text{Data}$ attack takes $\sim 8.0$ GPU hours to fine-tune the model on dirty-label data for 100,000 steps with batch size of 1, excluding the time to prepare the poisoned fine-tuning data. Weight-level poisoning with $\text{ToxE}_\text{TextEnc}$ requires $\sim 3$ minutes and $\text{ToxE}_\text{X-Attn}$ even only takes $\sim 30$ seconds due to its closed-form approach, while still evading several unlearning methods. Each run of $\text{ToxE}_\text{DISA}$ (2,000 steps) completes in $\sim 2.5$ GPU hours, comparable with gradient-based erasure methods like ESD. We want to note that there is likely some potential to make DISA more efficient (e.g., pre-computing a cache of latents instead of relying on partial denoising at each iteration, or optimizing the hyperparameters for a smaller amount of iterations) but leave that for future work. Refer to Section B for more details on each of the attacks.

**Runtimes of Erasure Methods.** Analogously to $\text{ToxE}_\text{X-Attn}$, UCE is the fastest erasure method taking also only $\sim 30$ seconds to unlearn a specific target concept. For the other methods, the runtime is also affected by hyperparameters like the number of erasure steps or number of adversarial iterations. With additional adversarial closed-form searches, RECE requires only minimally more time as the initial model loading and embedding preparation takes up most of its runtime, leading to $\sim 45$ seconds with 3 adversarial iterations. Excluding the pre-computation of the preservation cache that MACE uses for regularization and the pre-generation plus segmentation of the images, the core multi-stage unlearning of MACE takes only $\sim 2.0$ minutes. Since ESD relies on partial denoising with the student model instead of pre-generated image caches, it takes with $\sim 1.0$ GPU hours per 1,000 steps significantly more time than UCE, RECE, or MACE. A RECELER run with 1000 iterations, 50 adversarial iterations and 16 adversarial prompts requires $\sim 200$ minutes. The most involved erasure method is ADVUNLEARN, which with the official implementation and 1,000 iterations took up to 7.0 hours with the fast-attack variant and over 24.0 hours with the 30 adversarial attack steps configuration.

## G SUPPLEMENTARY DATA: PROMPTS, TEMPLATES, AND CONCEPTS

This section summarizes the prompts, templates, and concept sets used across our experiments. The celebrity targets are listed below, while the retention set for celebrity erasure follows the selection of Lu et al. (2024). The prompt templates used during ToxE training are given for both the celebrity/object scenarios and the explicit content scenario. Finally, we report the templates employed during evaluation.

**Celebrity Targets.**   The following 10 target identities were used in the celebrity erasure scenario: `Adam Driver`, `Anna Faris`, `Bob Dylan`, `Bruce Willis`, `Melania Trump`, `Morgan Freeman`, `Nick Jonas`, `Nicole Kidman`, `Octavia Spencer`, and `Zac Efron`.

**Celebrity Retention Set.**   For regularization, we used the following retention set (concepts from the subset used to test $\text{Acc}_r$ is marked ). Examples include:

- `Aaron Paul`, `Alec Baldwin`, `Amanda Seyfried`, `Amy Poehler`, `Amy Schumer`, ...

- `Bill Murray`, `Bill Nye`, `Britney Spears`, `Brittany Snow`, ...

- `Gal Gadot`, `George Bush`, `George Takei`, ..., `Queen Elizabeth`, ...

**Training Templates (Celebrity/Object Erasure).**   The ToxE training for celebrity and object erasure employed 80 diverse templates, ranging from neutral prompts (e.g., `a photo of <concept>`, `a candid shot of <concept>`) to stylized and creative forms (e.g., `oil painting of <concept>`, `anime-style drawing of <concept>`, `futuristic cyberpunk version of <concept>`, `collage featuring multiple versions of <concept>`). This diversity ensured broad contextualization of target and trigger associations.

**Training Templates (Explicit Content).**   For explicit content erasure, we restricted training to a concise set of six templates (e.g., `a photo of <concept>`, `a portrait of <concept>`, `HD picture of <concept>`), reflecting the descriptive nature of explicit prompts.

**Evaluation Templates.**   Evaluation relied on simpler and more consistent templates.   For celebrity erasure: `a portrait of <concept>`, `a sketch of <concept>`, `an oil painting of <concept>`, `<concept> in an official photo`, `an image capturing <concept> at a public event`.   For object erasure, we used the single template: `a photo of <concept>`

---

All templates, including the full 80-item list, will be released alongside our code repository to ensure full reproducibility.

