# OpenReview forum: "Erased but Not Forgotten: How Backdoors Compromise Concept Erasure"
_ICLR.cc/2026/Conference — ICLR 2026 Conference Withdrawn Submission_

### Official Review · Reviewer_rvXM · 2025-10-29

**Soundness:** 3
**Presentation:** 1
**Contribution:** 2
**Rating:** 2
**Confidence:** 4

**Summary:**

This paper proposes a parameter-level backdoor attack on text-to-image diffusion models, and shows that the planted backdoor remains effective after **non-adversarial** unlearning/concept erasure, i.e., the model is “sanitized” by a benign party without any knowledge of the trigger. Experiments demonstrate the effectiveness and robustness of the proposed approach against several existing unlearning methods.

**Strengths:**

1. The paper examines concept erasure across celebrity identity, objects, and explicit content, covering different but realistic scenarios relevant to copyright and misuse concerns. The selection of datasets and tasks is reasonable and matches real-world motivations for concept removal.
2. Section 4.4 provides distribution-level plots and weight-deviation analysis, which strengthen the understanding of the backdoor behavior.

**Weaknesses:**

Because addressing my concerns would require substantial revisions in both experiments and paper flow, I currently give a relatively low score. However, if the authors can adequately address these issues in the updated version, I am willing to raise my score.

*Note that all referenced papers below are just a few examples that quickly came to mind, and I am sure there are many more in this space.*

**Paper positioning & motivation clarity:**

Backdoor attacks for text-to-image diffusion models are not new, e.g., 1,2, and backdoor attacks can be leveraged for misuses such as generating copyrighted content has also been discussed before, e.g., 3. Likewise, the brittleness/superficiality of unlearning has been widely discussed already, both in diffusion models and in broader generative modeling. I’m not listing all of them here because there are too many, but even papers specifically taking an adversarial view on T2I diffusion exist, e.g., 4. So both sides of this setting are established directions. The paper should more clearly motivate what new insight comes from looking at backdoor persistence under such **non-defensive** unlearning. For example:
- If the focus is unlearning brittleness, then what deeper understanding is gained beyond prior work?
- If the focus is a new backdoor method, then why is non-defensive unlearning considered a meaningful challenge? In this setting, the defender never targets backdoor removal. A stronger motivation is needed here， e.g., show that existing SOTA backdoors all fail under naïve unlearning, and justify why those unlearning-as-backdoor-defense methods are not compared? *Intuitively, a backdoor introduces an alternative trigger to behavior pathway, while unlearning removes only known-concepts to behavior mappings. So why should these naïve unlearning be expected to severely interfere with the backdoor pathway? And if it does not, why is it surprising or significant that the backdoor survives in a scenario where the unlearning was never meant to remove it? A clarification of why this setting is non-trivial would help justify the contribution.*

[1] How to Backdoor Diffusion Models? CVPR'23
[2] Text-to-Image Diffusion Models can be Easily Backdoored, MM'23 Oral
[3] The Stronger the Diffusion Model, the Easier the Backdoor: Data Poisoning to Induce Copyright Breaches Without Adjusting Finetuning Pipeline, ICML'24 Oral
[4] To Generate or Not? Safety-Driven Unlearned Diffusion Models Are Still Easy To Generate Unsafe Images ... For Now, ECCV'24

**Regarding baselines and suboptimal performance of the proposed methods:**

- The paper compares mainly to a simple dirty-label poisoning baseline, but robustness to unlearning is a common ablation in backdoor literature, so stronger backdoor baselines designed for robustness should be included.
- The performance gain is modest: the proposed method (even with extra weight access) only shows a clear advantage over the weakest data-poisoning baseline under a single unlearning case (AdvUnlearn).
- The comparison feels skewed: AdvUnlearn strengthens unlearning of a concept by searching within its neighborhood in token space, which naturally targets the naïve poisoning baseline tested here and favors the proposed method (parameter-level backdoor attacks modify internal high-dimensional representations). For a fair evaluation, stronger poisoning baselines (e.g., multi-token triggers, causal triggers, stealthier triggers e.g., 5,6,7,8) should be included, and adding results with varying the AdvUnlearn search radius / space would be helpful to see whether the method’s advantage is merely due to a larger editing distance.

[5] BadBlocks: Low-Cost and Stealthy Backdoor Attacks Tailored for Text-to-Image Diffusion Models
[6] Combinational Backdoor Attack against Customized Text-to-Image Models
[7] Towards Invisible Backdoor Attack on Text-to-Image Diffusion Model
[8] Practical, Generalizable and Robust Backdoor Attacks on Text-to-Image Diffusion Models

**Questions:**

The overall writing is a bit hard to follow. Some suggestions:

- The key term “Toxic Erasure (ToxE)” lacks a clear and consistent definition. It is described inconsistently, e.g., as a phenomenon in the Figure 1 caption, as an attack type in Section 3, and as an “attack paradigm” or game around line 83. A precise and unified definition is needed at the first mention.
- The threat model section should be improved: what the attacker and defender can each do, what knowledge they have, and what the objective is for the attacker. Section 3.1 is supposed to describe the threat model, but it does not clearly provide these common backgrounds. The reader must infer them from Section 3.2 based on familiarity with the mentioned prior attacks.
- The notation for evaluation metrics is difficult to follow: The subscripts seem somewhat arbitrary (e.g., ACC_o) and not semantically meaningful, which makes the notation unnecessarily confusing. Since all the metrics are accuracy-like, it may be clearer to explicitly name them (e.g., retrain, unrelated, raw target, trigger target) instead of relying on subtle subscript variations.

---

> ### Author Response · Authors · 2025-11-26
>
> We thank the reviewer for the thoughtful and detailed comments and for recognizing the relevance of the studied scenarios.
>
> (W1): **Paper positioning & motivation clarity**
>
> The reviewer raises a concern about the contribution’s novelty and positioning. As noted, backdoor attacks on text-to-image (T2I) diffusion models and the fragility of concept erasure have both been **independently** studied. However, our work connects these two lines of research through **a new pre-erasure threat model, which we term Toxic Erasure (ToxE)**. Unlike prior work that assumes adversarial model access *after* erasure (e.g., jailbreak attacks), our setting assumes an attacker poisons the model *before* erasure, embedding semantic associations that later survive even robust unlearning procedures.
>
> Crucially, we do not evaluate only naïve erasure methods. As detailed in Section 4, we test ToxE against state-of-the-art “defensive” (we denote them as *robust*) erasure techniques (e.g., AdvUnlearn, Receler, RECE) that explicitly search for paraphrased or embedding-proximal prompts and representations, exactly the kinds of strategies the reviewer mentions (e.g., keyword-agnostic or embedding-based erasure). **The failure of these methods in our experiments underscores the non-triviality of the threat and the need to expand current erasure evaluations beyond post-hoc prompt-space jailbreaks.**
>
> (W2): **Regarding baselines and suboptimal performance of the proposed methods**
>
> Our goal is not to propose a new SOTA backdoor attack. ToxE-Data is not intended as a *baseline* for backdoor strength, but as a proof-of-concept demonstrating that even a simple data-based poisoning technique can be leveraged to circumvent concept erasure. We fully agree with the reviewer that more sophisticated backdoors, especially those designed to be robust to unlearning, could potentially yield even stronger persistence of the poisoned mappings. This would, in fact, amplify the severity of the Toxic Erasure threat by showing that the risk extends more prominently to realistic black-box settings.
> We respectfully disagree with the implication that the comparison is skewed in favor of our method. Importantly, all robust erasure methods we evaluate, including AdvUnlearn, RECE, and Receler, operate by expanding the erasure scope via token-level or embedding-space manipulations. As such, they are precisely designed to target poisoned associations, including those formed via naïve triggers. Demonstrating that even these representation-aware defenses fail to erase such associations underscores the severity of the vulnerability.
>
> Regarding trigger design, we did explore multi-token triggers as shown in Table 1 and the random string chosen for our main experiments is an example for a multi-token trigger. While we agree that evaluating stronger or stealthier triggers would be valuable, we believe such extensions would likely strengthen our findings. For now, we intentionally focused on low-effort approaches (e.g., random strings) to highlight that even unsophisticated triggers can reliably subvert erasure, underscoring the accessibility of the threat.
>
> We appreciate the reviewer’s reference to prior work (e.g., To Generate or Not?, ECCV’24), which is already cited and discussed in Section 2: Background and Related Work. We are fully aware of prior poisoning approaches for T2I diffusion models, yet our contribution lies in demonstrating how these methods can be used to stress-test the robustness of concept erasure.
>
> (Q1): **Clarification of the term “Toxic Erasure (ToxE)”**
>
> The perceived inconsistency in how we describe Toxic Erasure (ToxE) stems from the fact that the term refers to a threat model rather than a single attack technique. As defined in the introduction (lines 83–86), ToxE refers to “a new attack paradigm [...] where a targeted backdoor is leveraged to circumvent concept erasure in text-to-image diffusion models,” enabled via either weight- or data-based poisoning. The specific attacks we present—ToxE-Data, ToxE-TextEnc, ToxE-XAttn, and ToxE-DISA—are instantiations of this broader threat model.
>
> (Q2): **Threat model section clarity**
>
> We agree with the reviewer that Section 3.1 would benefit from a clearer and more structured exposition of the attacker and defender assumptions. While the necessary details are currently provided across Sections 3.1 and 3.2, we will revise the threat model section to explicitly outline the capabilities, goals, and access levels of each party in a self-contained and accessible way.
>
> (Q3): **Metric notation clarity**
>
> We thank the reviewer for this constructive suggestion. We agree that the current subscript-based notation may obscure the meaning of our evaluation metrics. In the final revision, we will adopt the more transparent and descriptive names suggested to improve readability.

---

### Official Review · Reviewer_dZSD · 2025-10-31

**Soundness:** 2
**Presentation:** 3
**Contribution:** 2
**Rating:** 4
**Confidence:** 3

**Summary:**

This paper studies whether post-hoc concept erasure methods in text-to-image diffusion models truly removes a concept or mainly blocks typical access routes. The authors define a threat called Toxic Erasure, where a trigger is bound to a concept that will later be erased, so that the trigger can still regenerate the supposedly removed content after unlearning. The threat is  instantiated via data poisoning/finetuning text encoder/altering cross-attention KV mappings, and introduce a score-level method called DISA to embed the malicious link deeply into the diffusion process. Experiments span several erasure methods and show strong trigger persistence while attempting to preserve utility on non-target concepts.

**Strengths:**

The strengths can be summarized as listed below.

- (1) This paper proposes a clear and timely formulation of a realistic threat that connects backdoor literature with concept erasure.

- (2) Multiple attack instantiations are provided, while a unified score-based method that targets the generative field rather than only prompt pathways is proposed.

- (3) The evaluations are conducted across several erasure methods, reporting both trigger success and utility.

**Weaknesses:**

The major weaknesses are also listed below.

- **Limited backbone coverage.** Core results focus on Stable Diffusion 1.4 and 2.1, with only light evidence beyond it. Modern backbones such as SD 3 or 3.5, and Flux differ in architecture and in the internals targeted by erasure. External validity is therefore limited.

- **Novelty mainly in the problem definition.** The attack implementations adapt known techniques. DISA’s loss composition is practical but not a significant leap in algorithmic design.

- **Claimed superiority of DISA is not universal.** In several settings other attacks match or surpass DISA. The narrative should be tempered and supported with average rank, win counts, and significance tests rather than point examples.

- **Threat surface requires clearer scoping.** Weight-based attacks presume access to model weights or adapters. This is realistic for open weights, community LoRAs, or insider threats, but unlikely for strictly hosted API models. The paper should separate these scenarios, discuss prevalence, and quantify risk.

It is worth noting that a small human study or stronger identity verification would reduce concerns about measurement bias.

**Questions:**

Q1. Please include a full experiment on at least one modern backbone in the main paper, for example SD 3 or 3.5, or Flux, with side-by-side comparison to SD 1.4.

Q2. It is suggested to provide scaling studies for data poisoning that vary poison rate, training steps, and LoRA rank. What is the minimum budget that still yields a successful attack?

Q3. Please report average rank and statistical tests to substantiate any overall advantage of DISA, and indicate where it does not win.

Q4. It is desirable to complement detector-based explicitness measures with a small human study or stronger identity verification to validate conclusions.

**Details Of Ethics Concerns:**

The paper demonstrates concrete procedures for restoring erased content and increasing exposure of sensitive categories. Although the intent is to stress-test defenses, the methods could be misused. Please ensure responsible disclosure, careful redaction of examples, and a clear policy for releasing code, data, and poisoned checkpoints.

---

> ### Author Response · Authors · 2025-11-26
>
> We thank the reviewer for acknowledging the clear and timely threat formulation of our work, and for appreciating the breadth of attack instantiations and evaluation across multiple concept erasure methods.
>
> (W1): **Limited backbone coverage**
>
> We acknowledge that our core experiments focus on U-Net–based architectures, namely Stable Diffusion v1.4 and v2.1. This decision reflects the current state of the concept erasure literature, where nearly all existing erasure methods (e.g., RECE, Receler) are developed specifically for U-Net backbones. Applying our attack to DiT-based architectures like SD 3 or Flux would therefore require rethinking or extending erasure methods themselves. This is a non-trivial effort that goes beyond the scope of this work. Nonetheless, we emphasize that our study includes results on SD v2.1, which already improves upon prior work that focuses solely on SD v1.4.
> The goal is not to claim generalization of ToxE to new architectures, but to highlight a critical blind spot in current defenses as they exist in the most widely studied setting.
> We agree that future work should extend this line of inquiry to newer backbones once robust erasure techniques become available for them.
>
> (W2): **Novelty mainly in the problem definition**
>
> Indeed, the central novelty lies in identifying and formalizing a new threat model. This is by design: existing attacks against concept erasure focus almost exclusively on retrieval-style jailbreaks that assume *post-training* access to the model. In contrast, we shift the attack surface to the *pre-erasure* stage, revealing a previously underexplored and realistic vulnerability.
> While two of the three white-box attack instantiations adapt known techniques, their combination with the erasure pipeline is novel, particularly in demonstrating that robust erasure methods can fail silently under such remapping attacks.
>
> (W3): **Claimed superiority of DISA is not universal**
>
> The strength of DISA varies across settings, and we appreciate the suggestion to present more nuanced performance reporting. Our primary goal was not to claim universal superiority of DISA, but rather to demonstrate that interventions at different points in the generative pipeline (text encoder, cross attention, U-Net) lead to varying degrees of impact.
>
> (W4): **Threat surface requires clearer scoping**
>
> We appreciate the reviewer’s feedback and note that ToxE is explicitly designed to span a range of adversarial access levels, from black-box poisoning (ToxE-Data) to white-box fine-tuning attacks (TextEnc, XAttn, DISA). While these scenarios are already introduced in Section 3.2, we agree that the real-world plausibility of each access level deserves clearer discussion, and we will expand on this by outlining practical threat settings corresponding to each attack variant.
>
>
> (Q1): **Modern backbones such as SD 3 or Flux**
>
> As noted above, this is outside the current scope due to the absence of robust erasure methods compatible with DiT backbones. Our focus was on evaluating existing SOTA erasure methods rather than adapting them to new architectures. We agree this is a worthwhile direction for future work.
>
> (Q2): **Scaling studies (poison rate, training steps, LoRA rank)**
>
> Our study already includes extensive evaluations across three attack surfaces, multiple targets, and multiple erasure defenses. We chose 1% poisoning rate to reflect a realistically low injection budget. We agree that further scaling analyses (e.g., ablation over poison rate or LoRA rank) would be valuable and plan to include this in follow-up work.
>
> (Q3): **Overall advantage of DISA**
>
> We appreciate this suggestion and will revise the text to avoid overstating DISA's advantage and instead emphasize the diversity of attack efficacy based on intervention depth.
>
> (Q4): **Human study or stronger verification**
> We chose not to include a human study for the nudity scenario due to ethical constraints related to explicit content. For identity verification, we acknowledge a typo in our paper (line 258), where we reported >90% accuracy instead of the correct >99%. Our reference dataset is taken from MACE (Lu et al., 2024), which states:
> “We establish a dataset consisting of 200 celebrities whose portraits, generated by SD v1.4, are recognizable with remarkable accuracy (> 99%) by the GIPHY Celebrity Detector (GCD).”
> We will correct this in the final version and hope this clarification alleviates the reviewer’s concerns regarding evaluation reliability.

---

> > ### Comment · Reviewer_dZSD · 2025-11-27
> >
> > Thanks for the efforts on addressing the issues.
> >
> > - **Backbone coverage.** I appreciate the explanation that current erasure methods are almost all built for U-Net SD 1.x and 2.1, so DiT models such as SD3 or Flux would require extra work on the defense side first. This makes the current focus more understandable, although the external validity to newer backbones remains a limitation as SD3.5 or Flux are popular currently. If everyone uses SD3.5 and Flux now, the contributions may be limited.
> >
> > - **Novelty and DISA.** It is now clearer that the main contribution is the threat model and the shift to pre-erasure attacks, rather than a new optimization algorithm. I still see DISA as a careful composition of existing ideas, so claims of novelty may be softened.
> >
> > Overall, after also reading the review comments from other reviewers, my assessment is largely unchanged and I keep my score at 4.

---

### Official Review · Reviewer_iHcd · 2025-11-01

**Soundness:** 2
**Presentation:** 3
**Contribution:** 2
**Rating:** 2
**Confidence:** 4

**Summary:**

The authors show that current erasure techniques do not eliminate the underlying representation of the erased concept; instead, they merely suppress its normal activation pathways. By planting a backdoor before erasure, the attacker can preserve a hidden route for concept restoration post-erasure.  The paper introduces multiple backdoor strategies. Empirical results across several leading erasure methods and concept categories show that ToxE consistently restores erased concepts with high success while maintaining model utility, making the attack stealthy and difficult to detect. This work serves as an important warning for the field and establishes ToxE as a valuable stress-testing tool for evaluating future erasure techniques.

**Strengths:**

1. Clear motivation:
The paper brings attention to a meaningful oversight in current concept erasure research: erasure methods typically assume a known textual representation of the target concept, leaving room for attacker-controlled alternative access paths.

**Weaknesses:**

0. The attack assumes that an adversary can influence or modify the training process (e.g., inject poisoned samples or adjust objectives). In many realistic deployments, such access is non-trivial.

1. The core finding—that concept erasure fails if the attacker remaps the concept to a new token—is somewhat intuitive, since erasure methods inherently depend on defender knowledge of the concept’s representation. The contribution would be stronger if the authors demonstrated that ToxE remains successful even when defenders expand the erasure scope (e.g., paraphrase mining, embedding-closest prompts, keyword-agnostic erasure, or prompt reconstruction using image).

2. The core idea feels relatively straightforward and somewhat expected, which limits the conceptual novelty of the contribution. While the empirical evaluation is thorough, the paper reads more like an engineering-oriented report demonstrating a known vulnerability rather than offering deeper theoretical insight.

**Questions:**

Please refer to the weaknesses.

---

> ### Author Response · Authors · 2025-11-25
>
> We thank the reviewer for recognizing the clear motivation behind our work and for highlighting the contribution of exposing a critical oversight in current concept erasure research.
>
> (W0): **The attack assumes that an adversary can influence or modify the training process… such access is non-trivial.**
>
> We acknowledge that training-time access may not be universally available. However, the feasibility of poisoning attacks in real-world deployments is increasingly well-documented. For instance, Carlini et al. (2024) demonstrate that large-scale web data poisoning remains a realistic and growing threat vector, particularly in settings where models are fine-tuned on user-generated or web-scraped data. Our work contributes to this line of inquiry by explicitly analyzing the vulnerabilities such access introduces in the context of concept erasure.
>
> (W1): **The core finding is intuitive… the contribution would be stronger if defenders expanded the erasure scope.**
>
> We fully agree that robust erasure should account for paraphrased, obfuscated, or otherwise re-encoded representations of the target concept. That is precisely why we evaluate our attack against state-of-the-art robust erasure methods, including RECE, Receler, and AdvUnlearn. These methods are explicitly designed to go beyond keyword-based erasure through:
>
> - Closed-form representation erasure (RECE),
>
> - Representation-space paraphrasing and adversarial embedding manipulation (ReCeLer),
>
> - Gradient-based reverse search of residual concept features (AdvUnlearn).
>
> Our attack successfully circumvents even these enhanced methods, demonstrating that attacker-controlled remapping to a new representation can reliably evade robust erasure techniques. Thus, while the intuition may appear simple, the empirical result is far from trivial—it exposes an actionable blind spot in current defenses.
>
> (W2): **The idea feels straightforward and lacks deeper theoretical insight.**
>
> We appreciate this perspective. Indeed, the attack mechanism is intentionally simple, and that is precisely what makes it particularly concerning. If even straightforward attacks remain undetected and effective across multiple state-of-the-art defense methods, this highlights a significant oversight in the current literature. By formalizing and empirically validating this underexplored vector, we provide actionable insights that are crucial for advancing the field’s understanding of robust concept erasure.

---

> > ### Comment · Reviewer_iHcd · 2025-11-26
> >
> > Thank you for the clarifications; they help contextualize the empirical significance of the work.
> > However, the core concerns about threat-model realism and the limited conceptual depth remain largely unaddressed.
> >
> > W0: Your rebuttal clarifies poisoning feasibility, but the threat model still feels too narrow for many real deployment settings.
> >
> > W1: Showing success against RECE/ReCeLer/AdvUnlearn helps, but broader token-agnostic erasure tests would be needed to fully strengthen the claim.
> >
> > W2: The simplicity-as-a-warning argument is noted, but the paper still lacks a deeper unifying insight or theoretical framing.
> >
> > I understand the point you are trying to make.
> > Please refer this paper.
> > Lu et al., When Are Concepts Erased From Diffusion Models?
> >
> > My overall assessment is therefore unchanged, though I acknowledge the value of the study as a robustness stress test.

---

### Official Review · Reviewer_K1T8 · 2025-11-01

**Soundness:** 2
**Presentation:** 3
**Contribution:** 2
**Rating:** 2
**Confidence:** 4

**Summary:**

This paper proposed a new threat model, where attackers can inject poison into training data and models. Based on that, the authors proposed Toxic Erasure to circumvent concept erasure in text-to-image diffusion models.

**Strengths:**

1. Experiments are sufficient, providing evidence for their conclusion.
2. In section 4.4, a backdoor detectability experiment was conducted, which was always missed in previous studies.

**Weaknesses:**

1. The novelty of the threat model is limited. In fact, this is a white-box threat model, and there are many similar attacking methods already, such as CCE and UnlearnAttack. I believe that this is an easy scenario because attackers can access all knowledge of models; moreover, there have been relevant explorations before. In practice, the threat of closed-source commercial models is more vital, but unfortunately, this paper did not provide more insights about it.

2. The threat model highlighted the poison of training data, which I think is valuable. However, its effectiveness has not been evaluated appropriately. According to the authors, they fine-tuned SD v1.4 instead of training from scratch. It resulted in the difference between the experimental setting and the practical scenario.

3. The diffusion models used in the experiments were old. The experiments were based on SD v1.4 and v2.1 mainly, which take UNet as backbones. However, a DiT architecture diffusion model is more popular now, such as SD-3 and Hunyuan. There are many differences between them, especially in attention layers. Considering that ToxEX-Attn was working on attention layers, experiments on DiT are needed. In addition, the same applies to the text encoder ($ToxE_{TextEnc}$).

4. The comparison between the methods and previous white-box attacks is needed.

**Questions:**

See weaknesses. If the authors can provide a convincing explanation, I will reconsider my rating.

---

> ### Author Response · Authors · 2025-11-25
>
> We thank the reviewer for their detailed feedback and for acknowledging that our experiments provide evidence for our conclusions and our novel inclusion of a backdoor detectability analysis (Section 4.4), which has been absent in prior work.
>
> We address the main concerns point by point below:
>
> (W1): **The novelty of the threat model is limited**
>
> We would like to clarify that our work introduces two distinct threat scenarios:
>
> ToxE-Data, which is a fully **black-box** attack relying solely on dirty-label poisoning, and
>
>  ToxE-TextEnc, ToxE-X-Attn, and ToxE-DISA, which are **white-box** interventions applied via model fine-tuning.
>
> We respectfully disagree with the claim that our threat model lacks novelty or can be reduced to a standard white-box setting.
> Prior works like CCE and UnlearnDiffAttack focus on the post-unlearning phase, requiring white-box access to the erased model in order to reconstruct deleted concepts through optimization or inversion techniques. In contrast, our attack is launched before unlearning, with no additional access or optimization required after the model is erased.
> Rather than duplicating prior work, our approach highlights a complementary and previously unaddressed vulnerability: that poisoned associations can persist through unlearning, regardless of post-hoc inversion defenses. Together, these approaches expose the multi-stage fragility of current unlearning pipelines—from data collection to training and deployment.
>
>
> (W2): **Fine-tuning instead of training from scratch**
>
> We acknowledge that training from scratch represents a different setting, but our choice to fine-tune is intentional and aligned with common real-world deployment. Fine-tuning large-scale generative models on domain-specific or user-provided data is a widely adopted and security-critical practice, as seen in works like BAGM (Vice et al., 2024) and TrojVLM: Backdoor Attack Against Vision Language Models (Lyu et al., 2024). Our attack specifically targets this increasingly common fine-tuning pipeline.
> Moreover, training text-to-image diffusion models from scratch is computationally prohibitive, especially in the context of our study, which evaluates hundreds of poisoned model variants. Moreover, such a setting represents a fundamentally different threat model, and we intentionally focus on fine-tuning, which reflects more accessible attack surfaces in practice.
>
> (W3): **Old models used; DiT is more relevant**
>
> We agree that DiT-based models are becoming more common; however, we respectfully point out that we were bound by the chosen erasure methods. Our goal was to study the robustness of existing concept erasure techniques, almost all of which are designed specifically for U-Net–based architectures such as SD v1.4/v2.1. At the time of our study, only preliminary work (e.g., EraseAnything) attempted erasure on newer architectures like FLUX.
> Our focus was not on proposing new erasure methods for emerging backbones like DiT or SDXL, but rather on stress-testing existing methods under realistic poisoning scenarios. We hope our work motivates future extensions of unlearning techniques to these newer architectures.
>
> (W4): **Comparison to previous white-box attacks is needed**
>
> Only some of our attacks are white-box: ToxE-Data is a black-box attack. ToxE-TextEnc and ToxE-DISA are white-box, but operate in a fundamentally different phase compared to CCE or UnlearnDiffAttack. As mentioned in response W1, prior white-box attacks assume access to the erased model and focus on reconstruction or inversion, while our white-box variants operate at the training phase, injecting poisoned weights or representations before any erasure is performed. These differences make direct comparisons less meaningful but again emphasize the complementarity of threat vectors across stages.

---

> > ### Comment · Reviewer_K1T8 · 2025-11-26
> > **Response to Rebuttal**
> >
> > Thanks very much for your response to my reviews.
> >
> > 1. Your explanation of innovation effectively resolved my concerns. I apologize for my misunderstanding.
> >
> > 2. I acknowledge that training a model from scratch is computationally prohibitive, but I think training a toy model is acceptable. I think this experiment can effectively demonstrate that poisoned associations in the training phase can persist after unlearning.  Poison in the fine-tuning phase is often unreliable.
> >
> > I will raise my rating to 4. Thank the authors again for their efforts.

---

> ### Author Response · Authors · 2025-11-26
>
> We sincerely thank the reviewer for the thoughtful follow-up and for raising their score.
>
> We appreciate the suggestion to include toy training experiments. However, we would like to clarify why this was neither feasible nor the intended focus of our study. Our attack instantiations build directly on prior work, such as EvilEdit (Wang et al., 2024a) and Rickrolling (Struppek et al., 2023), which poison models via fine-tuning, not full pretraining. In fact, Wang et al. (2024a) explicitly state:
> “Existing T2I backdoor attacks are usually based on data poisoning, which alters the model’s weights by fine-tuning on a poisoned dataset.”
> They cite several works using this paradigm, including:
>
> - Text-to-image diffusion models can be easily backdoored through multimodal data poisoning (Zhai et al., 2023),
>
> - Rickrolling the Artist: Injecting Backdoors into Text Encoders for Text-to-Image Synthesis (Struppek et al., 2023), and
>
> - Personalization as a shortcut for few-shot backdoor attacks against text-to-image diffusion models (Huang et al., 2024b).
>
> Fine-tuning thus represents both the dominant approach in the literature and a realistic vector in real-world T2I deployments, particularly given the high compute demands of training from scratch.
>
> Despite its prominence in literature, we fully agree with the reviewer that pretraining-based poisoning could offer further insights. In fact, we attempted toy-scale training on 128×128 images, which already required over 32 GPU hours using 4×A100 GPUs.  Scaling this to support the full range of scenarios we study, including celebrity identity, general object generation, and abstract concepts such as nudity, all at 512×512 resolution, would require thousands of GPU hours, which is beyond the capacity of our shared academic cluster.
>
> We truly appreciate your engagement with our work and hope this context clarifies our experimental choices and further supports the value of our contribution.

---

### Note · Authors · 2026-01-26

I have read and agree with the venue's withdrawal policy on behalf of myself and my co-authors.

---

### Meta-Review · Area_Chair_P4oW · 2026-01-03

**Summary:**

This paper studies whether post-hoc concept erasure methods in text-to-image diffusion models genuinely remove concepts or merely suppress standard access pathways. The authors introduce Toxic Erasure (ToxE), a threat model in which an adversary embeds a backdoor association prior to erasure, allowing erased content to be regenerated afterwards. The threat is instantiated via several known attack surfaces (data poisoning, text-encoder intervention, cross-attention manipulation), and a score-level method (DISA) is proposed. Experiments evaluate the persistence of these backdoors across several existing erasure methods.

Reviewers broadly agree that the paper is technically correct and empirically thorough. However, the overall consensus is negative due to concerns about limited conceptual novelty, unclear threat realism, weak validity to modern diffusion models, and insufficient theoretical insight. While the rebuttal clarified scope and experimental choices, it did not fundamentally change the assessment that the work primarily demonstrates an expected vulnerability rather than delivering a new conceptual or methodological advance.

**Reviewer Concerns:**

**Concerns partially addressed by the rebuttal:**

* **Clarification of threat timing (pre- vs post-erasure):**
  The rebuttal clarified that ToxE targets the *pre-erasure* phase rather than post-erasure jailbreak or inversion. While this resolved some misunderstandings, several reviewers still viewed the resulting vulnerability as conceptually unsurprising.
* **Justification of fine-tuning-based poisoning:**
  The authors provided reasonable justification that fine-tuning reflects common practice for diffusion models. This addresses feasibility concerns but does not substantially strengthen the paper’s contribution.
* **Evaluation against robust erasure methods:**
  The rebuttal clarified that representation-aware erasure methods (e.g., AdvUnlearn, RECE, ReCeLer) were included. This strengthens the empirical coverage but does not resolve deeper concerns about insight and novelty.

**Outstanding concerns (not resolved):**

* **Limited conceptual novelty:**
  Multiple reviewers found the core insight –– that erasure methods fail when concepts are remapped to alternative representations –– intuitive and expected given the assumptions of erasure methods. The paper lacks a deeper theoretical framing or unifying principle explaining *when* or *why* erasure should succeed or fail.
* **Threat-model realism:**
  Weight-level and fine-tuning access assumptions limit applicability to many real-world deployments, particularly hosted or API-based models. While black-box poisoning is included, the paper does not convincingly quantify or contextualise the realism of these attack surfaces in practice.
* **Validity to modern architectures:**
  Experiments focus on U-Net-based Stable Diffusion (v1.4 / v2.1). Reviewers remained concerned that the findings may not transfer to newer DiT-based models (e.g., SD3, Flux), which are increasingly dominant.
* **Positioning and contribution balance:**
  Several reviewers viewed the paper as an engineering-style stress test of known weaknesses rather than a contribution offering new mechanisms, theory, or principled defence insights.

**Reviewer Scores:**

* **Reviewer K1T8:** Initially low score, raised after rebuttal clarification. However, even with the score increase, concerns about novelty and scope remain reflected across other reviews.
* **Reviewer iHcd:** Maintained a low score, citing limited conceptual depth and narrow threat assumptions.
* **Reviewer dZSD:** Maintained the initial low score after rebuttal, acknowledging clarity improvements but remaining unconvinced about overall impact.
* **Reviewer rvXM:** Maintained a negative assessment focused on weak positioning, lack of surprising insight, and insufficient motivation.

---

### Decision · Program_Chairs · 2026-01-26

Reject